# A General Comprehensive Evaluation Method for Cross-Scale Precipitation Forecasts

Bing Zhang[1], Mingjian Zeng[1]*, Anning Huang[2]*, Zhengkun Qin[3,4], Couhua Liu[5], Wenru Shi[1],

Xin Li[1], Kefeng Zhu[1], Chunlei Gu[2], Jialing Zhou[1]

1. Key Laboratory of Transportation Meteorology of China Meteorological Administration, Nanjing Joint Institute

for Atmospheric Sciences, Nanjing 210041, China

2. School of Atmospheric Sciences, Nanjing University, Nanjing 210023, China

3. School of Atmospheric Sciences, Nanjing University of Information Science & Technology, Nanjing 210044,

China

4. Joint Center of Data Assimilation Research and Applications, Nanjing University of Information Science &

Technology, Nanjing 210044, China

5. National Meteorological Centre, Beijing 100081, China

**Correspondence**: Mingjian Zeng (swordzmj@qq.com) and Anning Huang (anhuang@nju.edu.cn)

## Short summary

By directly analyzing the proximity of precipitation forecasts and observations, a precipitation forecast accuracy score (PAS) method was constructed. This method does not utilize traditional contingency table-based classification verification, can replace the threat score (TS), equitable threat score (ETS) and other skill score methods, and can be used to calculate the accuracy of numerical models or quantitative precipitation forecasts.

## **Abstract**

With the development of refined numerical forecasts, problems such as score distortion due to the division of precipitation thresholds in both traditional and improved scoring methods for precipitation forecasts and the increasing subjective risk arising from the scale setting of the neighbourhood spatial verification method have become increasingly prominent. To address these issues, a general comprehensive evaluation method (GCEM) is developed for cross-scale precipitation forecasts by directly analyzing the proximity of precipitation forecasts and observations in this study. In addition to the core indicator of the precipitation accuracy score (PAS), the GCEM system also includes score indices for insufficient precipitation forecasts, excessive precipitation forecasts, precipitation forecast biases and clear/rainy forecasts. The PAS does not distinguish the magnitude of precipitation and does not delimit the area of influence; it constitutes a fair scoring formula with objective performance and can be suitable for evaluating rainfall events such as general and extreme precipitation. The PAS can be used to calculate the accuracy of numerical models or quantitative precipitation forecasts, enabling the quantitative evaluation of the comprehensive capability of various refined precipitation forecasting products. Based on the GCEM, comparative experiments between the PAS and threat score (TS) are conducted for two typical precipitation weather processes. The results show that relative to the TS, the PAS better aligns with subjective expectations, indicating that the PAS is more reasonable than the TS. In the case of an extreme precipitation event in Henan, China, two high-resolution models were evaluated using the PAS, TS, and fraction skill score (FSS), verifying the evaluation ability of PAS scoring for predicting extreme precipitation events. In addition, other indices of the GCEM are utilized to analyze the range and extent of both insufficient and excessive forecasts of precipitation, as well as the precipitation

forecasting ability for different weather processes. These indices not only provide overall scores similar to those of the TS for individual cases but also support two-dimensional score distribution plots, which can comprehensively reflect the performance and characteristics of precipitation forecasts. Both theoretical and practical applications demonstrate that the GCEM exhibits distinct advantages and potential promotion and application value compared to the various mainstream precipitation forecast verification methods.

## 1. Introduction

Precipitation is one of the most important forecasting elements in weather forecasting (Bi et al., 2016; Han et al., 2023). Short duration heavy rainfall often leads to flooding and geological disasters, causing widespread and severe impacts (Zhong et al., 2022; Yang et al., 2023). Precipitation forecasts, as focuses and challenges in meteorological department operations, have drawn widespread attention from governments, societies and the public (Bi et al., 2016; Hao et al., 2023). Scientifically evaluating precipitation forecasts helps people gain a clear understanding of the current precipitation forecast levels and maintain appropriate psychological expectations for such forecasts. Moreover, such evaluations assist forecasters in rationally analysing the quality and characteristics of quantitative precipitation forecast systems and aid researchers in understanding the level, strengths and weaknesses of various types of forecasting systems, which in turn, offers valuable insights to improve these systems (Zhong et al., 2022; Zhang et al., 2022; Liu et al., 2022b; Gofa et al., 2018). However, there are several shortcomings in current precipitation verification approaches. For instance, with traditional scoring methods, small errors in the location or timing of convective features can lead to false alarms and missed events, and their utility is limited regarding diagnosing

model errors such as a displaced forecast feature or an incorrect mode of convective organization;
thus, traditional scoring methods often fail to reflect model performance improvements (Ahijevych et
al., 2009). For high-resolution precipitation forecasts, even if the spatial distribution and intensity of
precipitation are consistent with the observations, slight spatial and temporal deviations between
forecasts and observations may still result in a large false alarm ratio and missed alarm ratio, leading
to lower forecast scores (Zhao and Zhang, 2018). With the rapid development of seamless fine
quantitative precipitation forecasts, the need for objective and rational evaluations of the accuracy
and characteristics of precipitation forecasts has become increasingly important and urgent (Chen et
al., 2021).
Precipitation forecast verification involves various methods, including traditional contingency
table-based classification verification and spatial verification methods. The traditional verification
method can be traced back to 1884, when Finley introduced a dichotomous contingency table for
tornado forecasts and evaluated these forecasts using the proportion correct scoring method (Finley,
1884). Subsequently, systematic attention was given to the evaluation of forecast classification
methods, and Finley's forecast verification method became a classic example of the discussion of
forecast scoring methods (Murphy, 1996). Shortly thereafter, Gilbert (1984) proposed two scoring
methods, namely, the ratio of verification and the ratio of success in forecasting. The ratio of
verification later became known as the threat score (TS) (Palmer and Allen, 1949) or the critical
success index (Donaldson et al., 1975; Mason, 1989). The ratio of success is referred to as the
Gilbert skill score (GSS) (Schaefer, 1990) or the equitable threat score (ETS) (Doswell et al., 1990;
Gandin and Murphy, 1992). The TS encourages correct event forecasts (hits) and accounts for the
impacts on the false alarm and missed alarm ratios, which can better guide forecasters or research
and development personnel in making reasonable subjective and objective predictions compared to
relying solely on simple "accuracy". Meanwhile, the ETS eliminates the influence of random
forecasts on the score, resulting in a fairer skill score (Liu et al., 2022a).

In addition to the TS and ETS, the methods of traditional contingency table-based classification

verification include Peirce skill score (PSS) (Peirce, 1884; Hanssen and Kuipers, 1965; Murphy and
Daan, 1985; Flueck, 1987), Heidke skill score (HSS) (Doolittle, 1885; Doolittle, 1888; Heidke,
1926), probability of detection (POD), frequency bias (BIAS), accuracy (ACC), false alarm ratio
(FAR), missing ratio (MR), probability of false detection (POFD), etc. The PSS is a fair score index
that is equal to the hit rate minus the false detection probability; the HSS eliminates the influence of
random forecasts, and the results can reflect the forecast skill (Liu et al., 2022a). Many studies have
reviewed and compared these two scoring methods (Doswell et al., 1990; Schaefer, 1990; Marzban,
1998; Mason, 2003). In extreme weather event verification (including severe convective weather
such as short duration heavy rainfall), the traditional scoring methods (such as the TS and ETS) for
dichotomous events often yield scores of zero when the occurrence probability of the object being
verified is very low. Therefore, Stephenson proposed the extreme dependency score (EDS) for
evaluating extreme events. The EDS has the advantage that different forecast systems converge to
different values and has no explicit dependence on the bias of the prediction system (Stephenson et
al., 2008; Casati et al., 2008).

It has been more than a century since Gilbert proposed two scoring concepts, i.e., the ratio of

verification and the ratio of success in forecasting (later known as the TS and ETS). The TS and ETS
have been widely used for the performance evaluation of threshold-based event forecasts despite
their evident shortcomings (Stephenson et al., 2008). Today, in various forecast verification

applications, including high-resolution quantitative precipitation and extreme weather forecast verification, the TS and ETS remain mainstream approaches (Tang et al., 2017; Wei et al., 2019; Chen et al., 2021; Liu et al., 2023). With the continuous introduction of new scoring methods, several problems in traditional verification have been solved. However, the advantageous position of the TS remains unchallenged. The reasons for which, although varied, are worthy of attention, but include its objectivity and practicality.

The traditional TS categorizes precipitation according to thresholds and performs verification using a dichotomous contingency table. The TS can be viewed as a measure of forecast accuracy that excludes hit forecasts for "non-occurrence" precipitation events (referred to as no precipitation), and its calculation formula is simple, objective and standardized. However, there are two main limitations of the TS. First, precipitation is categorized by thresholds based on the contingency table, which has limitations in terms of classification. The drawback of artificially dividing precipitation into different threshold ranges is that it cannot guarantee that two adjacent precipitation values will always fall within the same threshold range. Slightly different precipitation values are not within the same threshold, which can lead to precipitation score distortion. The second limitation is related to the so-called "double penalty" issue. With the development of high-resolution numerical weather forecasting and the shortening of the spacing between model grid points, some medium- and small-scale phenomena have been portrayed by models. However, it is difficult for high-resolution numerical forecasts to match the characteristics of the observed medium- and small-scale forecasts, resulting in traditional scoring methods often cannot reflect these improvements in terms of model performance. Assuming a constant forecast area, when there is a small deviation in the timing and location of events between a forecast and an observation, both "false alarms" and "missed alarms"

will occur, which is referred to as the "double penalty" phenomenon. This phenomenon leads to a score lower than the subjectively expected result, making it difficult to obtain appropriate verification scores when a forecast that "looks good" is not as good as one that "looks bad" (Ahijevych et al., 2009; Wilks, 2006; Ebert, 2008; Chen et al., 2021). For low-probability events with limited sample size for verification, such as torrential rain and short-term heavy rainfall, the "double penalty" issue becomes more prominent. The TS and ETS for torrential rain are often at the unskilled end of the scoring values (Chen et al., 2019). In recent years, new mainstream scoring methods have addressed most of the abovementioned limitations but still have shortcomings. Such methods include the improved gradient decreasing method, which still results in poor scores for good forecasts, and the neighbourhood spatial verification method, which has too many subjective components and may miss medium and small scale information.

To address the limitations of threshold-based precipitation classification, and improve the verification effect, e.g., the gradient decreasing method (hereinafter referred to as the magnitude-improved TS) is used to verify the accuracy of rainstorm forecasts (Yang et al., 2017), and appropriate weights are assigned to close forecast values to avoid scores of zero (Table 1). However, the magnitude-improved TS still has limitations. For example, if the observed 24-hour accumulated precipitation is 50 mm, when forecast A is 48 mm and forecast B is 98 mm, forecast A is evidently better than forecast B. According to the original TS, forecast B scores 1 point, while forecast A does not score any points. For the magnitude-improved TS, forecast B scores 1 point, as it still falls within the same magnitude category as the observed precipitation, while forecast A scores only 0.4 points, which still fails to reflect the fact that forecast A is superior to forecast B (Table 2). By employing the new scoring method, i.e., the precipitation accuracy score (PAS), which will be discussed later,

forecast A scores 0.998 points, while forecast B scores 0.398 points, confirming the rationality and
validity of this new method.

To address the "double penalty" issue, a common approach is to employ the neighbourhood

spatial verification method (also known as the fuzzy method), which has two specific processing
forms. The first approach is simple upscaling, which uses a certain method (such as value averaging,
maximum, value weighting) to select values within the scale range, adjusting the high-resolution
forecast and observation information to a larger scale to reduce the accidental information of
high-resolution data, and then using the traditional skill score (Yates et al., 2006; Weygandt et al.,
2004). The other form is the improved neighbourhood spatial verification method proposed by
Roberts and Lean (2008). By referring to the Murphy skill score, this method obtains comprehensive
evaluation information by comparing the occurrence frequency (probability) of precipitation within
different scale windows. If the forecasted occurrence frequency closely approximates the observed
occurrence frequency, the forecast is considered valuable (Zhao and Zhang, 2018). From the
perspective of the precipitation occurrence probability within the analysis region, the precipitation
occurrence probability for observations and forecasts is the ratio of the precipitation area to the
analyzed area of the region, which is referred to as the fraction skill score (FSS). These two
processing methods effectively solve the "double penalty" problem, but neither can address the issue
of excessive smoothness of the precipitation fields during the upscaling process (Zhao and Zhang,
2018), which may result in the omission of some small- to medium-scale information (Zepeda-Arce
et al., 2000).

The neighbourhood spatial verification method considers values that are spatially and

temporally adjacent between forecasts and observations during the matching process, thus relaxing
the strict requirements for spatiotemporal matching (Ebert, 2008; Casati et al., 2008). However, since
the determination of the neighbourhood range is a rather subjective process, it hinders the
standardization of verification scores and lacks comparability, which may negatively affect objective
quantitative verification. Numerous experiments have shown that there is an obvious improvement in
the scoring values after adopting the neighbourhood spatial verification method (Chen et al., 2019),
particularly for forecasts of large-magnitude precipitation. Nevertheless, the purpose of scoring is not
to achieve a monotonous increase in scoring values but rather to follow the principle of objectivity as
much as possible. Errors are errors and cannot be solved by simply lowering the standard. Instead,
reasonable and fair criteria should be utilized to reflect the true extent of errors.
Currently, numerical weather forecasts and intelligent gridded forecasts have been developed to
output high-resolution precipitation products, while precipitation observations, whether in the form
of gridded or station data, are already high-resolution. Staying at the dichotomous classification level
for precipitation verification not only wastes existing data resources but also fails to meet the
evaluation requirements of refined forecasts. Therefore, to adapt to the development of refined
forecasts, a new scoring method is needed. In light of this, a comprehensive verification index for
precipitation forecasts is designed, and the following five aspects are considered. (1) The impact of
categorical events on rainstorm forecasts should be reduced. In particular, high-resolution forecasts
can refer to continuous variables for scoring methods. Especially for the evaluation of
high-resolution precipitation forecasts, the scoring method of continuous variables can be borrowed
for reference. (2) The design of the scoring method should aim to minimize subjective factors such as
the artificial range division and condition settings, ensuring scoring objectivity and comparability. (3)
The designed scoring performance indices should possess ideal attributes such as fairness, base rate
independence, suitability for extreme events, and boundedness as much as possible. (4) The devised
scoring method should be easy to promote, concise and efficient, with clear concepts and scientific
rationality. (5) Different comprehensive verification indices for precipitation forecasts should reflect
the forecasting performance and characteristics of high-resolution quantitative precipitation products
from various perspectives.
In this study, on the basis of analyzing the limitations of traditional verification methods as well
as improved methods, a new general comprehensive evaluation method (GCEM) for cross-scale
precipitation prediction is proposed. This method is applied and verified through practical examples.
The remainder of this paper is organized as follows. Section 2 provides an overview of various
scoring indices and their attributes in the GCEM and introduces the optimization processing method
for the PAS index in the application. Through ideal experiments, the characteristics of the scoring
methods are analyzed based on the score curves described in Section 3. Section 4 presents
comparative experiments, including the new scoring method, the traditional scoring method and the
neighbourhood spatial verification method based on typical cases. Finally, a summary and discussion
are presented in Section 5.
**2 Cross-scale general comprehensive evaluation method**
**2.1 Overview of the general comprehensive evaluation method**
To address the issues of "distorted scores due to the division of precipitation thresholds and
increased subjective risks brought about by the setting of the neighbourhood spatial verification
method" in traditional and improved precipitation scoring methods, referring to the verification
method for heavy rainfall forecasts based on predictability (Chen et al., 2019) and combining the
advantages of the relative and absolute errors in this study, a GCEM is constructed by directly
analysing the proximity of forecasted precipitation to observed precipitation. It primarily includes the
PAS, and the expression of its core scoring function is as follows:
$$PAS = \begin{cases} \sin\left(\frac{\pi}{2} \cdot \frac{x}{u}\right), & 0 \leq x < u \\ e^{-\left(\frac{x-u}{u}\right)^2}, & 0 < u \leq x \end{cases} \tag{1}$$
where PAS represents the scoring value, x is the forecasted precipitation (mm), and u is the observed
precipitation (mm). The PAS falls between 0 and 1, where a higher score indicates a better
precipitation forecast effect. When PAS = 1, it signifies a perfect forecast, indicating that the
forecasted and observed precipitation match entirely. For Eq. (1), given the observation value u>0
mm, when the forecasted precipitation is 0 mm, then PAS=0, indicating that the model has no
forecast skill. When the forecasted precipitation amount is sufficiently large, PAS → 0, indicating no
forecast skill as well (Fig. 1). Additionally, considering the large fluctuation characteristics of the
function curve when the observed precipitation is less than 10 mm, Eq. (1) was smoothed and
optimized (see Section 2.2 for details).
The GCEM system also includes the following indices:
(1) Insufficient precipitation index (IPI), whose core scoring function expression is as follows:
$$IPI = \sin\left(\frac{\pi}{2} \cdot \frac{x}{u}\right) - 1, \ 0 \leq x < u \tag{2}$$
where IPI represents the scoring value, reflecting the degree of underestimation in precipitation
forecasts when the forecasted value is less than the observed value. The IPI falls within [−1, 0),
where a value closer to 0 indicates a lower degree of underestimation.
(2) Excessive precipitation index (EPI), whose core scoring function expression is as follows:
$$EPI = 1 - e^{-\left(\frac{x-u}{u}\right)^2}, \quad 0 < u < x \tag{3}$$
where EPI represents the scoring value, reflecting the degree of overestimation in precipitation
forecasts when the forecasted value exceeds the observed value. The EPI falls within (0, 1), where a
value closer to 0 indicates a lower degree of overestimation.

(3) Insufficient and excessive precipitation index (IEPI), whose core scoring function

expression is as follows:
$$\text{IEPI} = \begin{cases} \sin\left(\frac{\pi}{2} \cdot \frac{x}{u}\right) - 1, & 0 \le x < u \\ 1 - e^{-\left(\frac{x-u}{u}\right)^2}, & 0 < u \le x \end{cases} \quad (4)$$

where IEPI represents the scoring value, reflecting the degree of deviation of the forecasted
precipitation from the observed precipitation. The IEPI falls within $[-1, 1)$, where a value closer to 0
indicates a lower degree of forecast deviation. An IEPI less (more) than 0 indicates an insufficient
(excessive) forecast, and an IEPI equal to 0 represents an unbiased forecast.

Additional explanation: Eqs. (2-4) are a series of theoretical indicator formulas derived from Eq.

(1), therefore Eqs. (2-4) are referred as the core calculation formulas for the IPI, EPI, and IEPI,
respectively. In practical applications, the optimized solution will be used (see Section 2.2) to
calculate the IPI, EPI, and IEPI for the situations of $u \ge 0.1$ mm or $x \ge 0.1$ mm.

(4) The PAS clear/rainy forecast accuracy score (PASC), whose scoring function expression is

as follows.
$$\text{PASC} = \begin{cases} 1 & 0 \le u < 0.1 \text{ and } 0 \le x < 0.1 \\ \text{PAS}_{|ux0.1} & u \ge 0.1 \text{ or } x \ge 0.1 \end{cases} \quad (5)$$

where PASC represents the PAS scoring value for clear/rainy forecasts. "$0 \le u < 0.1$ and $0 \le x <$
$0.1$" denotes the correctly forecasted non-precipitation event with PASC=1. $\text{PAS}_{|ux0.1}$ denotes the
overall PAS for precipitation forecasts under specific conditions where the observed precipitation $u \ge$
$0.1$ mm or the forecasted precipitation $x \ge 0.1$ mm.

The discussion below pertains to the characteristics of the PAS scoring method. As an ideal performance indicator, the PAS has the attributes of boundedness, fairness, sensitivity disparity, suitability for extreme events and moderate symmetry.

(1) Boundedness. The PAS scoring values range between 0 and 1. A PAS score of 1 represents an ideal forecast, while a score of 0 indicates that there is observed precipitation but no forecasted precipitation or that the forecasted precipitation is sufficiently large. The scoring range is consistent with that of traditional TS, making it easy to compare and evaluate the scoring methods and suitable for practical forecast verification applications.

(2) Fairness. The PAS scoring method constitutes a scoring formula in an objective form without a subjective boundary definition. Precipitation forecasts are verified without magnitude or delimitation of the area of influence, and the closer to the observed situation the forecast is, the higher the score, which is fair.

(3) Sensitivity disparity. According to the Chinese national standard GB/T 28592—2012 "Grade of precipitation" on the classification of precipitation grades, the public is more sensitive to low-grade precipitation forecasts. As rainfall intensity increases, the public's sensitivity gradually decreases; that is, the public has a higher tolerance for errors in response to heavier rainfall forecasts. In other words, large errors in the forecasts of heavy rainfall events may be considered equivalent to smaller errors in weaker rainfall events in terms of forecast scoring. As shown in Fig. 1, the intersection point on the PAS scoring curves for the observed precipitation amounts of 25 mm and 100 mm corresponds to a forecasted amount of 42.4 mm. That is, the forecast errors are 17.4 mm and 57.6 mm for the observed 24-hour accumulated precipitation amounts of 25 mm and 100 mm, respectively, while the scores are both 0.62. From the perspective of forecast service effectiveness,

this aligns with general public perception.

(4) Suitability for extreme events. From the PAS scoring curves for forecasts corresponding to

different observed precipitation amounts (u = 10, 25, 50 and 100 mm) (Fig. 1), it is evident that the
PAS scoring method performs well in evaluating precipitation event forecasts at the level of
torrential rain and above. For example, when the observed precipitation is 100 mm, with forecasted
amounts of 59 mm and 147.2 mm, the PASs are both 0.8, whereas the TSs are 0 and 1, and the
improved TSs are 0.8 and 1, respectively. This result indicates that the PAS is suitable for scoring
heavy rainfall events, meeting the general applicability requirements as a scoring method that does
not degrade due to extreme events.

(5) Moderate symmetry. In Eq. (1), let the observed precipitation is the independent variable u,

and the forecasted precipitation is the parameter x. Similarly, for different magnitudes of forecasted
precipitation (parameter x = 10, 25, 50 and 100 mm) and observed precipitation (variable u) ranging
from 0 to 300 mm, the corresponding scores are shown in Fig. 2. The scores also vary with the
degree of proximity between forecasts and observations. Figures1 and 2 exhibit similar trends but are
not identical, illustrating that the PAS possesses moderate symmetry.
## 2.2 PAS verification for precipitation forecasts

From the properties of the core verification function of the PAS, it is noted that when the

observed precipitation $u$ < 10 mm, there is a large gradient in the PAS curve. A slight change in the
forecasted value ($x$) can result in a large fluctuation in the PAS. To account for this characteristic,
based on a comprehensive analysis in combination with the sensitivity of forecasters and the public
to small-scale precipitation, a smoothing optimization scheme is applied to the PAS curve for
accumulated precipitation below 10 mm. Similarly, the IPI, EPI, IEPI and PASC curves are
appropriately smoothed and optimized according to their respective definitions.
Assumptions:
(1) PAS $= 0.6\mathrm{PAS}_{|u\to 0}$ when $u = 0$ mm, and $x \neq 0$ mm;
$\mathrm{PAS}_{|u\to 0}$ denotes the PAS for the case of observed precipitation $0 < u \leq 0.1$ mm;
(2) PAS $= 0.6\mathrm{PAS}_{|x\to 0}$ when $x = 0$ mm, and $0 < u < 10$ mm;
$\mathrm{PAS}_{|x\to 0}$ denotes the PAS for the case of forecasted precipitation $0 < x \leq 0.1$ mm.
1. When the observed precipitation $u = 0$ mm and the forecasted precipitation $x > 0$ mm (Fig.
3a), let PAS $= 0.6\mathrm{PAS}_{|u\to 0}$, then,
$$\mathrm{PAS} = 0.6e^{-\left(\frac{x}{10}\right)^2} \qquad x > 0 \tag{6}$$
2. When the forecasted precipitation $x = 0$ mm and the observed precipitation $0 < u < 10$ mm
(Fig. 3b), let PAS $= 0.6\mathrm{PAS}_{|x\to 0}$, then,
$$\mathrm{PAS} = 0.6\sin\left(\frac{\pi}{2}\cdot\frac{10-u}{10}\right), \quad 0 < u < 10 \tag{7}$$
The coefficient was set to 0.6. According to Eqs. (6-7), when the situation is the observation
u=0 mm and forecast x=0.1 mm or the observation u=0.1 mm and forecast x=0 mm, PAS=0.6,
suggesting that the forecast effect has just reached the standard, like when the ACC reaches 0.6,
which indicates that the model forecast effect is available (Zhao and Zhang, 2018).
3. When the observed precipitation $0 < u < 10$ mm and the forecasted precipitation $x \neq 0$ (Fig.
3c), then,
$$\mathrm{PAS} = \begin{cases} \sin\left(\frac{\pi}{2}\cdot\frac{x-u+10}{10}\right), & 0 < x < u, \ 0 < u < 10 \\ e^{-\left(\frac{x-u}{10}\right)^2}, & u \leq x, \quad 0 < u < 10 \end{cases} \tag{8}$$
4. When the observed precipitation $u \geq 10$ mm (Fig. 3d), then

$$\text{326} \qquad \text{PAS} = \begin{cases} \sin\left(\dfrac{\pi}{2} \cdot \dfrac{x}{u}\right), & 0 \le x < u, \ \ u \ge 10 \\[3mm] e^{-\left(\frac{x-u}{u}\right)^2}, & u \le x, \ \ u \ge 10 \end{cases} \qquad (9)$$

To compare with the traditional scoring method, the new scoring method for precipitation
forecasting adopts the "classification before verification, no classification during verification"
approach. Scoring for precipitation processes over different accumulation periods is referenced but
not limited to the commonly used precipitation classification approaches in practical operations, as
shown in Tables 3-5.

## 3 Ideal experimental validation of the new verification method

### 3.1 Validation of forecast scoring results for general precipitation

General precipitation refers to precipitation ranging from light rain to heavy rain, i.e., 24-hour
accumulated precipitation within [0.1 mm, 50 mm). Figure 4 shows the schematic diagram of PAS
scores for general precipitation. The forecasted amounts are compared under conditions when the
24-hour accumulated precipitation is 10 mm, 25 mm and 45 mm and the PAS scores are 0.8, 0.7, 0.5
and 0.3 (Table 6). When the observed precipitation is 10 mm, the forecasted amounts of 5.9 mm and
14.7 mm both have a PAS score of 0.8, with differences from the perfect forecast value (10 mm) of
4.1 mm and 4.7 mm, respectively; the forecasted amounts with a PAS score of 0.3 are 1.9 and 21.0
mm, differing by 8.1 mm and 11.0 mm from the perfect forecast value (10 mm), respectively. When
the observed precipitation is 25 mm, the forecasted amounts with a PAS score of 0.8 are 14.7 mm
and 36.8 mm, with differences from the perfect forecast value (25 mm) of 10.3 mm and 11.8 mm,
respectively; the forecasts with a PAS score of 0.5 are 8.3 mm and 45.8 mm, differing by 16.7 mm
and 20.8 mm from the perfect forecast value (25 mm), respectively.
For forecasts with the same observed precipitation and the same scores, the absolute errors of an

insufficient forecast and observation are smaller than those of an excessive forecast and observation, and the higher the scores are, the closer the absolute errors of the forecasts. When the observed precipitation is 50 mm, only the insufficient precipitation forecast is scored since a precipitation forecast exceeding 50 mm is not considered within the scope of general precipitation evaluation. The scoring experimental results align with expectations.

## 3.2 Validation of forecast scoring results for precipitation at the level of torrential rain and above

Figure 5 shows a schematic diagram of the PASs when the amount of precipitation exceeds the storm magnitude. The predicted precipitation is compared when the 24-hour cumulative observed precipitation is 25 mm, 50 mm, and 100 mm with PAS scores of 0.877, 0.7, 0.5, 0.3, and 0.1 (Table 7). When the observed precipitation is 25 mm, only forecasts ≥ 50 mm are involved in the rating, with PASs of 0.3 and 0.1 for forecasts of 52.4 and 62.9 mm, respectively.

When the PAS is 0.877 and the observed precipitation is 50 mm, the predicted values are 34.1 and 68.1 mm, respectively; when the observed precipitation is 100 mm, the predicted values are 68.1 and 136.2 mm, respectively. When the observed precipitation is 50 or 100 mm, the prediction is 68.1 mm, with a score of 0.877. The absolute error is 18.1 mm for the excessive precipitation forecast and 31.9 mm for the insufficient precipitation forecast. This result indicates that the scoring tolerance increases as the grade of observed precipitation increases and gradually expands through continuous changes, avoiding discontinuous increases caused by changes in magnitude.

When the observed precipitation is 50 mm and the PAS is 0.3, the insufficient forecast is 9.7 mm and the excessive forecast is 104.9 mm. When the observed precipitation is 100 mm, the predictions for a PAS of 0.3 are 19.4 and 209.7 mm, respectively. When the observed precipitation is

50 mm, the insufficient forecast with a PAS of 0.1 is 3.2 mm, and the excessive forecast is 125.9 mm. When the observed precipitation is 100 mm, the predictions with a PAS of 0.1 are 6.1 and 251.7 mm, respectively.

Under constant observed precipitation conditions, for forecasts with the same score, the absolute error between the insufficient forecast and the observed precipitation is smaller than that between the excessive forecast and the observed precipitation. The higher the score is, the smaller the absolute error between the forecast and the observation. Moreover, the scoring tolerance increases with increasing observed precipitation. The scoring experimental results conform to expectations.

## 4 Example-based comparative experiments for the new verification method

Different examples are selected for the new precipitation verification method, and its multifaceted characteristics are demonstrated through comparative experiments. In Section 4.1, two typical cases are selected, the performance characteristics of the PAS and TS are compared, and the indicators of insufficient and excessive forecasts and spatial verification in the GCEM are analyzed. In Section 4.2, typical case of extreme precipitation event is selected, and the forecast results of different high-resolution models using the PAS, TS, and FSS methods are evaluated to verify the advantages and characteristics of the new precipitation verification method for extreme precipitation events.

### 4.1 Comparative experiments of two typical processes

**4.1.1 Introduction of typical cases**

Comparative experiments of PAS and traditional TS are conducted for 12-hour accumulated precipitation for two typical cases. One case pertains to the precipitation weather process occurring from 00:00 to 12:00 UTC on 16 July 2019 (referred to as "Case 1"), which is dominated by a weak

weather system. The other case relates to the precipitation weather process occurring during 00:00 to
12:00 UTC on 13 June 2020 (referred to as "Case 2"), which is predominantly associated with a
strong weather system.

Both precipitation cases are associated with precipitation during the Meiyu period. Case 1,

which occurred during the Meiyu period of 2019 and was characterized by scattered precipitation
under weak synoptic-scale forcing. The low-intensity shear line system is located south of the
Yangtze River. There are two precipitation concentration areas, one at the intersection of Hunan
Province and Jiangxi Province and the other covering the majority of Zhejiang Province. The
precipitation process in Case 2 (12–13 June) was the first round of widespread rainstorms during the
Meiyu period of 2020, including heavy precipitation affected by a low-level vortex shear system.
The western section of the low-level vortex shear is relatively stable, while the eastern section
slightly presses southwards. Southwesterly airflow developed and pushed northwards, and a strong
wind speed belt persisted for a long time in the Jianghuai region. Moreover, the Jianghan–Jianghuai
region maintained a high-energy and high-moisture state, resulting in persistent heavy rainfall.

A subjective analysis of these two weather processes reveals that for the event on 16 July 2019

(Fig. 6), the forecasted precipitation intensity and rainfall areas are relatively consistent with the
observations. There are two distinct heavy rainfall areas in the eastern and southern parts of the
Yangtze River, with particularly high accuracy in forecasting scattered rainstorms in Zhejiang
Province located in the eastern section. In contrast, for the precipitation weather process on 13 June
2020 (Fig. 7), it is evident that there is an overestimation of the precipitation forecast.
**4.1.2 Data and methods**

The observed precipitation data are provided by the China Meteorological Administration

multisource merged precipitation analysis system (CMPAS), developed by the National Meteorological Information Centre of China. The CMPAS integrates hourly precipitation data from nearly 40,000 automatic meteorological stations in China and provides radar-based quantitative precipitation estimation and satellite-retrieved precipitation products with a spatial resolution of $0.05° × 0.05°$. The predicted precipitation data with 3 km resolution are from the Precision Weather Analysis and Forecasting System (PWAFS) model, a regional refined forecast model, developed by the Jiangsu Provincial Meteorological Bureau. These data are output once per hour.

The specific methods are as follows.

(1) Determine the verification domain and verification points. The verification domain covers the Huang–Huai region of China (28°N-38°N, 111°E-123°E). The verification points are defined based on the grid points of the observed precipitation data, their spatial resolution is $0.05° × 0.05°$, and the total number of verification grid points is 48,000 (200 × 240).

(2) Prepare the observed and forecasted precipitation data and interpolate the forecasted precipitation data onto the observed grid points. The observed 12-hour accumulated precipitation data are derived by accumulating the hourly precipitation data from the CMPAS. The forecasted 12-hour accumulated precipitation data are obtained by subtracting the zero-field data from the 12-hour forecast field data. Since the grid points of the observed and forecasted precipitation data do not coincide and the grid spacing is small, the nearest neighbour method is used in this study to match the forecasted data to the grid points of the observed precipitation. Specifically, the forecasted data on the model grid nearest to the observed grid are used as the forecasted value at this observed grid.

(3) Analyze the relationship between the forecasted precipitation and observed precipitation.

The scores for each verification grid point and the overall scores for each verification area are
calculated based on the scoring formula for each index in the GCEM system. Then, the verification
result file is generated in NetCDF format. On this basis, distribution maps for the scores of various
indices in the GCEM system are produced. Additionally, the total TS and clear/rainy TS for different
precipitation magnitudes within the verification area are calculated based on the TS and clear/rainy
TS formulas.
**4.1.3 Analysis of the comparative experiment results**
For the precipitation process on 16 July 2019, the traditional TSs for different rainfall
categories, such as clear/rainy and 12-hour accumulated precipitation $\geq$ 0.1 mm, $\geq$ 10 mm, $\geq$ 25 mm
and $\geq$ 50 mm, are all lower than the traditional TSs for the weather process on 13 June 2020. For
example, the TS is 0.381 for 12-hour accumulated precipitation $\geq$ 0.1 mm during 00:00 to 12:00
UTC on 16 July 2019 (Table 8), while this score is 0.625 for that during 00:00 to 12:00 UTC on 13
June 2020 (Table 9), which differs from the subjective judgement.
For the precipitation process during 00:00 to 12:00 UTC on 16 July 2019 , the PASs for
clear/rainy and 12-hour accumulated precipitation $\geq$ 0.1 mm, $\geq$ 10 mm and $\geq$ 25 mm are all higher
than those for the precipitation process during 00:00 to 12:00 UTC on 13 June 2020 . For instance,
the overall PAS is 0.617 for 12-hour accumulated precipitation $\geq$ 0.1 mm during 00:00 to 12:00 UTC
on 16 July 2019. This PAS is higher than the PAS of 0.457 for the precipitation process during 00:00
to 12:00 UTC on 13 June 2020 , which aligns with subjective judgement.
For the precipitation process during 00:00 to 12:00 UTC on 16 July 2019, the PAS for each
magnitude is higher than the corresponding TS, addressing the issue of TSs being lower. For the
precipitation process during 00:00 to 12:00 UTC on 13 June 2020, the PASs for clear/rainy and the
magnitudes of ≥ 0.1 mm and ≥ 10 mm are lower than the corresponding TSs, whereas the PASs for
the magnitudes of ≥ 25 mm and ≥ 50 mm are higher than the corresponding TSs. This result indicates
that the PAS is different from the magnitude-improved TS and the neighbourhood spatial verification
method. Both the magnitude-improved TS and the neighbourhood spatial verification method
increase the tolerance, leading to a monotonous increase in scores. This result also demonstrates that
the PAS has good discrimination ability for extreme events. The PAS assigns scores based on the
proximity of the forecast to the observation, making it more reliable for precipitation evaluation than
the TS.
**4.1.4 Analysis of the indices in the new verification method**
Modern forecast verification is based mainly on spatial verification methods to compensate for
the shortcomings of traditional methods. The literature review of Gilleland et al. (2009) defines four
main categories of methods: neighbourhood, scale separation, features based, and field deformation
(Ahijevych et al., 2009). These methods can analyze more comprehensively in specific individual
cases, but seem to be less able to provide direct overall scoring results than traditional scoring
methods in the statistics of long time series. GCEM is based on point-to-point scoring statistics,
without a radius of influence, no isolation of features at each scale, and no definition of objects in the
forecast and observation to analyze the similarity of the objects or to fit the forecast objects through
deformation operations. However, the GCEM still has spatial attributes that can discriminate spatial
forecast characteristics (e.g., insufficient or excessive forecasting scenarios) for different categories
of precipitation, and the GCEM can carry out statistical verification of long time series and produce
overall scoring results.
Regarding the issue of analyzing the sources of errors from the verification results, objectively
tracing these errors back from a single score can only determine whether an error was "insufficient
(missed alarm)" or "excessive (false alarm)". However, the advantage of the GCEM lies in its ability
to decompose the score for each verification point and examine the forecasting performance at each
point, which is different from the dichotomous evaluation approach with only 0 and 1 outputs. These
indices not only provide overall scores for individual cases similar to the TS but also offer
two-dimensional score distribution plots, which can comprehensively reflect the performance and
characteristics of precipitation forecasts.
Figure 8 shows the distributions of the 12-hour accumulated precipitation PASC scores. In these
two cases, due to the high accuracy of non-precipitation forecasts, the overall PASC scores are
relatively high. However, for Case 1, the scores in Zhejiang are lower and scattered within a small
area. In contrast, for Case 2, there is a large area occupying most of the Jianghuai region with low
scores. Therefore, the PASC score of Case 1 (0.808) is higher than that of Case 2 (0.734).
Figure 9 shows the PAS distributions of 12-hour accumulated precipitation with magnitudes of
$\geq 0.1$ mm, $\geq 10$ mm and $\geq 25$ mm. The blank points in the figure are the points that are excluded in
the scoring, following the scoring principle of "classification before verification, no classification
during verification" described in Section 2. From the PAS distributions of different magnitudes, for
Case 1, the high and low scores in the Zhejiang region are scattered among them. In contrast, for
Case 2, the scoring areas in the Jianghuai region have a larger area of low scores than high scores.
Therefore, Case 1 has higher PASs for the three categories ($\geq 0.1$ mm, $\geq 10$ mm and $\geq 25$ mm) than
Case 2, and the distributions also allow for distinguishing the areas with better and worse forecasting
performance.
Figure 10 shows the IPI, EPI and IEPI distributions of 12-hour accumulated precipitation. In

terms of the IPI, for Case 1, the large-value IPI areas are located at the intersection of Anhui, Zhejiang and Jiangxi, in the Hunan-Jiangxi region, as well as in the southern part of Hebei. For Case 2, the large-value IPI areas are situated along the Yangtze River in Anhui and Jiangxi, as well as at the intersection of Henan and Shanxi. The IPIs for Case 1 and Case 2 are -0.376 and -0.400, respectively, indicating that Case 2 shows a slightly higher level of insufficient forecasts (Table 10). In terms of the EPI, for Case 1, the large-value EPI areas are in Zhejiang and Jiangxi. In contrast, for Case 2, the large-value EPI areas are located in most of Hunan, Hubei, Anhui and Jiangsu, exhibiting a wide southwest–northeast orientation with a large area and degree. The EPI for Case 2 is larger than that for Case 1. The IEPI is a comprehensive reflection of under- and over- precipitation, and its value reflects the degree of insufficient and excessive precipitation forecasts. From the distributions of insufficient and excessive precipitation forecasts in Case 1, it is evident that the insufficient and excessive forecasts are roughly equivalent, with an IEPI of 0.057. However, for Case 2, the distribution of the excessive forecasts is obviously larger than that of the insufficient forecasts, with an IEPI of 0.325. This result indicates that Case 2 has poorer forecasting performance, with larger excessive forecasts being an important factor.

Consequently, analyzing the locations of insufficient and excessive precipitation forecasts from the figures in conjunction with the characteristics of the forecasting process can provide useful insights for improving forecasts.

## 4.2 Comparison experiment of extreme rainfall events

### 4.2.1 Introduction of the "21.7" extreme rainstorm event in Henan, China

From 17 to 23 July 2021, a rare extreme rainstorm event occurred in Henan Province, China. The extremely heavy rainstorm started in the southeastern of Henan Province in the morning of 17

July, then extended to the northern region, and ended in the morning of 23 July, lasting more than 6
days. The rainstorm occurred against the background of typhoon, Huang-Huai vortex, shear line and
convergence line, and was caused by the coupling of the low-level jet and boundary layer jet,
combined with the uplift of terrain (Wang et al., 2022; Su et al., 2021; Shi et al., 2021).

The period from 00:00 UTC on 18 July to 00:00 UTC on 22 July 2021 is the concentrated

period of heavy precipitation. To facilitate the study, the heavy rainstorm process is divided into three
periods: (1) 00:00 UTC on 19 July - 00:00 UTC on 20 July 2021. (2) 00:00 UTC on 20 July - 00:00
UTC on 21 July 2021. (3) 00:00 UTC on 21 July - 00:00 UTC on 22 July 2021. (Figs. 11a-c).
**4.2.2 Data and methods**

The observed precipitation data are provided by the CMPAS, with a spatial resolution of 0.05° ×

0.05°, similar to the case in Section 4.1. The forecast data come from two models. One is the PWAFS
model, which has a horizontal resolution of 3 km, similar to the case in Section 4.1. The other is the
global-regional assessment and prediction system (GRAPES) model independently developed by the
China Meteorological Administration, which has a horizontal resolution of 3 km.

(1) Determine the verification domain and verification points. The verification domain covers

the region of (30°N-40°N, 107.5°E-117.5°E). The verification points are defined based on the grid
points of the observed precipitation data, their spatial resolution is 0.05° × 0.05°, and the total
number of verification grid points is 40,401 (201 × 201).

(2) Prepare the observed and forecasted precipitation data and interpolate the forecasted

precipitation data onto the observed grid points. The 24-hour cumulative precipitation observation
data of the three periods were obtained from the 24-hour precipitation data of the CMPAS. The
forecast precipitation data in the three periods are the cumulative precipitation with a forecast time of
12 to 36 hours. For the case described in Section 4.1, the nearest neighbour method is used to match
the forecast data to the grid points of the observed precipitation.

(3) Analyze the relationship between the forecasted precipitation and observed precipitation.

PAS, TS and FSS were compared for the extreme rainstorm event in Henan, China.

As mentioned earlier, the FSS belongs to the neighbourhood category of spatial verification

methods and is an advanced evaluation method widely used in recent years. It can still yield valuable
scores when the model prediction intensity is spatially biased and can also represent the scale
information of forecasting skills. Therefore, in this case, the FSS scoring method was added for
comparative experiments. For FSS verification, 15 km, 25 km, 45 km, 75 km and 120 km are used as
the neighbourhood distances.

The brief steps of FSS calculation are as follows: 1. Determine the domain scope. Set the

neighbourhood point n, such as when n=3 (n is odd), the neighbourhood range is 15 km × 15 km, 2.
Calculate the spatial density in the observed binary observation fields (Eq.10), 3. Calculate the
spatial density in the binary forecast fields(Eq.11), 4. Calculate $FSS_{(n)}$ (Eq.12). (Please refer to the
article of Roberts and Lean (2008) for details.)
$$O(n)(i,j) = \frac{1}{n^2}\sum_{k=1}^{n}\sum_{i=1}^{n} I_o\left[i+k-1-\frac{n-1}{2}, j+l-1-\frac{n-1}{2}\right] \quad (10)$$
$$M(n)(i,j) = \frac{1}{n^2}\sum_{k=1}^{n}\sum_{i=1}^{n} I_M\left[i+k-1-\frac{n-1}{2}, j+l-1-\frac{n-1}{2}\right] \quad (11)$$
$$FSS_{(n)} = 1 - \frac{\frac{1}{N_xN_y}\sum_{i=1}^{N_x}\sum_{j=1}^{N_y}\left[O_{(n)i,j}-M_{(n)i,j}\right]^2}{\frac{1}{N_xN_y}\left[\sum_{i=1}^{N_x}\sum_{j=1}^{N_y}O_{(n)i,j}^2+\sum_{i=1}^{N_x}\sum_{j=1}^{N_y}M_{(n)i,j}^2\right]} \quad (12)$$
where $i$ ranges from 1 to $N_x$, $N_x$ is the number of columns in the domain and $j$ ranges from 1 to $N_y$,
$N_y$ is the number of rows. $I_o$ and $I_M$ are binary fields. $O_{(n)}(i, j)$ is the resultant field of observed
fractions for a square of length $n$. $M_{(n)}(i, j)$ is the resultant field of model forecast fractions obtained.
**4.2.3 Analysis of the comparative experiment results**
(1) Questionnaire survey of the effectiveness of model forecasting
Fifty-two questionnaires were completed by 32 researchers and 20 forecasters. The names of the
PWAFS and GRAPES models used for comparison were omitted and replaced with Model 1 and
Model 2, respectively.
The survey results show that 52 people believe that the forecasting effect of periods A and C of
Model 2 (GRAPES) is good, 19 people believe that the forecasting effect of period B of Model 1
(PWAFS) is good, and 33 people believe that the forecasting effect of period B of Model 2
(GRAPES) is good. Fifty-two people think that Model 2 (GRAPES) is good in general.
(2) Indices analysis and comparison between the two models
The high-resolution regional models used for evaluation are (1) PWAFS 3km and (2) GRAPES
3km, and the modelled precipitation is the accumulated precipitation of 24 hours in the forecast
12-36 hours. The evaluation results are as follows (Tables 11-16):
The results show that in this process, the evaluation results of different methods on the forecast
skill of the PWAFS and GRAPES models are basically consistent and in line with the subjective
evaluation statistical results. However, PAS scores have obvious advantages in the evaluation of
rainstorms and above, especially extreme rainstorms. It can be seen from the six rating scales that the
TS and FSS have almost no ability to evaluate precipitation above 250 mm, and the scores are
generally at the unskilled end of 0 and no more than 0.2 (Chen et al., 2019). The PAS scores can also
distinguish differences and provide different scores for situations where the forecasting effect is
good.
For example, when evaluating precipitation above 250 mm, the scores of TS for PWAFS in all
three periods are 0.000, and the scores of GRAPES in the three periods are 0.000, 0.045 and 0.044.
The scores of FSS (45 km) for the PWAFS in all three periods are 0.000, and the scores of GRAPES
in the three periods are 0.000, 0.218 and 0.137, respectively. This indicates that the TS and FSS (45
km) have little ability to assess the heavy rainfall of this process.
The PAS scores for PWAFS in the three periods are 0.229, 0.302 and 0.153, and those for
GRAPES in the three periods are 0.338, 0.637 and 0.528, indicating that PAS has the ability to
evaluate heavy rainstorms (above 250 mm) in this process. The evaluation results show that
GRAPES is superior to PWAFS in predicting heavy rainfall.
The evaluation capabilities of PAS, TS, and FSS for precipitation above 100 mm are further
analyzed. The scores of TS for the PWAFS (GRAPES) are 0.035, 0.257, and 0.042 (0.178, 0.451,
and 0.284) in the three periods, respectively. The scores of FSS (45 km) are 0.129, 0.550, and 0.103
(0.432, 0.767, and 0.613) for the PWAFS (GRAPES) in the three periods, respectively. The
evaluation effect of FSS (45 km) is better than that of TS. The evaluation feature of FSS is to
examine the predictability scale of the model to reflect its predictive ability; however, due to the
subjectivity of selecting neighbourhood scales, its score lacks comparability. While the PAS scores
are 0.246, 0.492 and 0.253 (0.573, 0.581 and 0.492) for the PWAFS (GRAPES) in the three periods,
it can be seen that the PAS also has a good ability to assess heavy rainstorms in this process.
In small-magnitude precipitation (above light and moderate rain) verification, the FSS scores
tend to approach 1 as the neighbourhood distance expands, making it difficult to compare forecast
differences between models. The PAS scores can also distinguish the differences in forecast
effectiveness for small-magnitude precipitation.
In conclusion, different scoring methods were used to evaluate the skill of different models to
predict extreme precipitation events in July 2021 in Henan, China, and the evaluation characteristics

of different scoring methods were indicated. The results show that the PAS scoring method has obvious advantages in the evaluation of extreme precipitation events and can also reflect the differences in the small magnitude precipitation forecasting effects of the models well compared to those of the TS and FSS methods.

## 5 Discussion and conclusion

By analyzing the advantages and disadvantages of the traditional TS, magnitude-improved TS and neighbourhood spatial verification methods, a new precipitation verification method, GCEM, was designed and constructed from the perspective of the proximity of the forecast to the observation. This method consists of the core indicator of the PAS, as well as multiple indicators such as IPI, EPI, IEPI and PASC.

The PAS index consists of sine and e-exponential functions. Additionally, considering the characteristics of large fluctuations in the function curves when observed precipitation is less than 10 mm, the formula has been smoothed for optimization. The PAS method adopts the principle of "classification before verification, no classification during verification", which can serve as an alternative to skill scores such as the TS and ETS for verifying quantitative precipitation forecasts. This method is characterized by objective and transparent rules and easy generalization. Moreover, this approach possesses attributes of an ideal precipitation scoring method, such as fairness, boundedness and moderate symmetry. Therefore, it can be used to calculate the accuracy of numerical models or quantitative precipitation forecasts, as well as evaluate the comprehensive forecasting capabilities of various refined quantitative precipitation forecast products. The GCEM can also evaluate the performance of numerical forecasts on clear/rain forecasts, as well as insufficient precipitation forecasts, excessive precipitation forecasts and precipitation forecast biases.

In addition to the overall score, two-dimensional score distribution maps can be generated for each index in the GCEM system. These maps offer a comprehensive reflection of the precipitation forecasting performance of the numerical models and serve as a reference for improving model forecasts.

This new verification method is validated based on the forecast scoring results for general precipitation and precipitation at the level of torrential rain and above, and the verification results align with expectations. Comparative experiments are also conducted on two typical processes using the new verification method. For Case 1, the subjective judgement is relatively good, but the TS is lower. Conversely, for Case 2, the subjective judgement is poorer, yet the TS is higher. Verification using the PAS reveals that forecasts with better subjective judgement receive higher scores, and forecasts with poorer subjective judgement receive lower scores. Therefore, PAS aligns with public expectations.

The PAS, TS and FSS methods were used to compare and verify the "21.7" extreme precipitation event in Henan, China, to reflect the evaluation characteristics of different scoring methods. The results show that the PAS scoring method can not only reflect the difference in the small-magnitude precipitation forecast effect of models, but also has obvious advantages in the evaluation of extreme precipitation events.

In addition, the National Meteorological Centre of China conducted long-term series large-scale sample testing on this method in 2023. Based on the ECMWF model's 24-hour and 48-hour precipitation forecasts from March 2022 to February 2023, the assessment results show that compared to the TS, the PAS is less affected by the randomness of the sample, and the relative size relationship of different time forecast scores is more stable.

From the construction of the GCEM to ideal experiments and case analysis, it is evident that this evaluation system, especially the PAS method, is a suitable method for quantitative precipitation evaluation. However, the PAS still has subjective flaws, such as the determination of coefficients in the PAS expression [0.6 in Eqs. (6) and (7)] when the observed or forecasted precipitation is 0 mm. Once these coefficients are determined, they apply to all precipitation scoring, thus becoming an objective component in practice.

***Code and data availability.*** The source code and data of this work can be found at https://www.doi.org/10.5281/zenodo.10784525 (Zhang et al., 2024). The readme file can be found at https://www.doi.org/10.5281/zenodo.10784525 (Zhang et al., 2024), which includes the compiling environment and steps to repeat this work, as well as other relevant content descriptions (code, data, output files, module code main interfaces, etc.).

***Author contributions.*** BZ designed the evaluation method, completed the experiments, and wrote the paper. MZ provided advice on the planning and application of the evaluation method. AH provided suggestions for the evaluation method and contributed to paper revisions, ZQ contributed to paper revisions, and CL provided long-term series large-scale sample comparison test results for the evaluation method. All authors discussed the results and commented on the paper.

***Competing interests.*** The contact author has declared that none of the authors has any competing interests.

***Disclaimer.*** Publisher's note: Copernicus Publications remains neutral with regard to jurisdictional claims made in the text, published maps, institutional affiliations, or any other geographical representation in this paper. While Copernicus Publications makes every effort to include appropriate place names, the final responsibility lies with the authors.

*Acknowledgements.* This study was supported by grid precipitation analysis data, forecasted
precipitation data and numerical computing capabilities provided by the Jiangsu Provincial
Meteorological Bureau of China, as well as long-term series large-scale sample testing conducted at
the National Meteorological Centre of China.
*Financial support.* This research is supported by the National Key Research and Development
Program of China under Grant 2021YFC3000904, the Beijige Open Research Fund under Grant
BJG202403 and the Jiangsu Collaborative Innovation Center for Climate Change.

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

**Table 1.** Gradient decrease scoring table for station-by-station (time) rainstorm forecasts. The values are

normalized, i.e., score = original data/100.

| Observation | Forecast (mm) | | | |
|---|---|---|---|---|
| (mm) | 25-49.9 | 50.0-99.9 | 100.0-249.9 | ≥250 |
| <25.0 | -- | 0 | 0 | 0 |
| 25.0-39.9 | -- | 0.4 | 0 | 0 |
| 40.0-49.9 | -- | 0.7 | 0.4 | 0 |
| 50.0-99.9 | 0.4 | 1 | 0.8 | 0.4 |
| 100.0-249.9 | 0 | 0.8 | 1 | 0.9 |
| ≥250.0 | 0 | 0.4 | 0.8 | 1 |


**Table 2.** Examples of station-specific rainstorm precipitation scoring.

| | Observation | Forecast A | Forecast B | Correct, Reasonable or False |
|---|---|---|---|---|
| Precipitation | 50 mm | 48 mm | 98 mm | -- |
| Forecast effect | -- | Good | Bad | Correct |
| Classic TS | -- | 0 | 1 | False |
| Improved TS | -- | 0.4 | 1 | False |
| PAS | -- | 0.998 | 0.398 | Reasonable |


**Table 3.** Classification of PAS for short-term heavy rainfall.

| Scoring name | Notes on the scoring application |
|---|---|
| $PAS_{\|ux10}$ | PAS score for 1-hour observed precipitation u ≥ 10 mm or forecasted precipitation x ≥ 10 mm |
| $PAS_{\|ux20}$ | PAS score for 1-hour observed precipitation u ≥ 20 mm or forecasted precipitation x ≥ 20 mm |



**Table 4.** Classification of PAS for 12-hour accumulated precipitation.

| Scoring name | Notes on the scoring application |
|---|---|
| PASC | 12-hour PAS clear and precipitation forecast accuracy score |
| | 12-hour PAS overall precipitation prediction verification score |
| $PAS_{\|ux0.1}$ | PAS score for observed precipitation u ≥ 0.1 mm or forecasted precipitation x ≥ 0.1 mm |
| $PAS_{\|ux10}$ | PAS score for 12-hour observed precipitation u ≥ 10 mm or forecasted precipitation x ≥ 10 mm |
| $PAS_{\|ux25}$ | PAS score for 12-hour observed precipitation u ≥ 25 mm or forecasted precipitation x ≥ 25 mm |
| $PAS_{\|ux50}$ | PAS score for 12-hour observed precipitation u ≥ 50 mm or forecasted precipitation x ≥ 50 mm |
| $PAS_{\|ux100}$ | PAS score for 12-hour observed precipitation u ≥ 100 mm or forecasted precipitation x ≥ 100 mm |







**Table 5.** Classification of PAS for 24-hour accumulated precipitation.

| Scoring name | Notes on the scoring application |
|---|---|
| PASC | 24-hour PAS clear and precipitation forecast accuracy score |
| $PAS_{|ux0.1}$ | 24-hour PAS overall precipitation prediction verification score<br>PAS score for observed precipitation u ≥ 0.1 mm or forecasted precipitation x ≥ 0.1 mm |
| $PAS_{|ux10}$ | PAS score for 24-hour observed precipitation u ≥ 10 mm or forecasted precipitation x ≥ 10 mm |
| $PAS_{|ux25}$ | PAS score for 24-hour observed precipitation u ≥ 25 mm or forecasted precipitation x ≥ 25 mm |
| $PAS_{|ux50}$ | PAS score for 24-hour observed precipitation u ≥ 50 mm or forecasted precipitation x ≥ 50 mm |
| $PAS_{|ux100}$ | PAS score for 24-hour observed precipitation u ≥ 100 mm or forecasted precipitation x ≥ 100 mm |


**Table 6.** Examples of forecast verification scores for general precipitation (u = 25, 50 and 100 mm).

| PAS value | Observation u=10 mm | | Observation u=25 mm | | Observation u=45 mm | Observation u=50 mm (No comparison) |
|---|---|---|---|---|---|---|
| | Insufficient forecast x | Excessive forecast x | Insufficient forecast x | Excessive forecast x | Insufficient forecast x | Insufficient forecast x |
| PAS=0.8 | 5.9 | 14.7 | 14.7 | 36.8 | 26.6 | 29.5 |
| PAS=0.7 | 4.9 | 16.0 | 12.3 | 39.9 | 22.2 | 24.7 |
| PAS=0.5 | 3.3 | 18.3 | 8.3 | 45.8 | 15.0 | 16.7 |
| PAS=0.3 | 1.9 | 21.0 | 4.8 | -- | 8.7 | 9.7 |




**Table 7.** Same as Table 6, but for precipitation at the level of torrential rain and above (u = 25, 50 and 100 mm).

| PAS value | Observation u=25 mm | Observation u=50 mm | | Observation u=100 mm | |
|---|---|---|---|---|---|
| | Excessive forecast x | Insufficient forecast x | Excessive forecast x | Insufficient forecast x | Excessive forecast x |
| PAS=0.877 | -- | 34.1 | 68.1 | 68.1 | 136.2 |
| PAS=0.7 | -- | 24.7 | 79.9 | 49.4 | 159.7 |
| PAS=0.5 | -- | 16.7 | 91.6 | 33.3 | 183.3 |
| PAS=0.3 | 52.4 | 9.7 | 104.9 | 19.4 | 209.7 |
| PAS=0.1 | 62.9 | 3.2 | 125.9 | 6.4 | 251.7 |


**Table 8.** PAS and TS of 12-hour accumulated precipitation from 00:00 to 12:00 UTC on 16 July 2019.

| | Clear/rainy | ≥0.1 mm | ≥10 mm | ≥25 mm | ≥50 mm |
|---|---|---|---|---|---|
| PAS | 0.808 | 0.617 | 0.256 | 0.200 | 0.104 |
| TS | 0.690 | 0.381 | 0.194 | 0.076 | 0.006 |


**Table 9.** Same as Table 8, but from 00:00 to 12:00 UTC on 13 June 2020.

| | Clear/rainy | ≥0.1 mm | ≥10 mm | ≥25 mm | ≥50 mm |
|---|---|---|---|---|---|
| PAS | 0.734 | 0.457 | 0.228 | 0.185 | 0.116 |
| TS | 0.816 | 0.625 | 0.338 | 0.149 | 0.036 |


**Table 10.** Accuracy indices of insufficient precipitation forecast (IPI), excessive precipitation forecast (EPI) and
insufficient and excessive precipitation forecast (IEPI) of 12-hour accumulated precipitation for two precipitation
processes.

| | IPI | EPI | IEPI |
|---|---|---|---|
| Case 1 | -0.376 | 0.389 | 0.057 |
| Case 2 | -0.400 | 0.597 | 0.325 |




**Table 11.** PAS, TS and FSS scores of PWAFS 24-hour accumulated precipitation from 00:00 UTC on 19 July to

00:00 UTC on 20 July 2021.

|  | clear/rainy | ≥0.1 mm | ≥10 mm | ≥25 mm | ≥50 mm | ≥100 mm | ≥250 mm |
|---|---|---|---|---|---|---|---|
| PAS | 0.598 | 0.487 | 0.301 | 0.256 | 0.254 | 0.246 | 0.229 |
| TS | 0.823 | 0.774 | 0.377 | 0.229 | 0.115 | 0.035 | 0.000 |
| FSS(15 km) | --- | 0.909 | 0.637 | 0.452 | 0.259 | 0.090 | 0.000 |
| FSS(25 km) | --- | 0.923 | 0.680 | 0.486 | 0.281 | 0.102 | 0.000 |
| FSS(45 km) | --- | 0.939 | 0.732 | 0.526 | 0.307 | 0.129 | 0.000 |
| FSS(75 km) | --- | 0.953 | 0.778 | 0.559 | 0.335 | 0.180 | 0.003 |
| FSS(120 km) | --- | 0.964 | 0.820 | 0.592 | 0.365 | 0.226 | 0.007 |


**Table 12.** PAS, TS and FSS scores of PWAFS 24-hour accumulated precipitation from 00:00 UTC on 20 July to
00:00 UTC on 21 July 2021.

|  | clear/rainy | ≥0.1 mm | ≥10 mm | ≥25 mm | ≥50 mm | ≥100 mm | ≥250 mm |
|---|---|---|---|---|---|---|---|
| PAS | 0.653 | 0.578 | 0.408 | 0.398 | 0.427 | 0.492 | 0.302 |
| TS | 0.789 | 0.743 | 0.500 | 0.429 | 0.434 | 0.257 | 0.000 |
| FSS(15 km) | --- | 0.891 | 0.750 | 0.690 | 0.687 | 0.475 | 0.000 |
| FSS(25 km) | --- | 0.908 | 0.789 | 0.731 | 0.725 | 0.507 | 0.000 |
| FSS(45 km) | --- | 0.928 | 0.837 | 0.782 | 0.771 | 0.550 | 0.000 |
| FSS(75 km) | --- | 0.945 | 0.878 | 0.831 | 0.815 | 0.598 | 0.003 |
| FSS(120 km) | --- | 0.958 | 0.912 | 0.877 | 0.858 | 0.654 | 0.042 |


**Table 13.** PAS, TS and FSS scores of PWAFS 24-hour accumulated precipitation from 00:00 UTC on 21 July to
00:00 UTC on 22 July 2021.

|  | clear/rainy | ≥0.1 mm | ≥10 mm | ≥25 mm | ≥50 mm | ≥100 mm | ≥250 mm |
|---|---|---|---|---|---|---|---|
| PAS | 0.656 | 0.533 | 0.346 | 0.322 | 0.296 | 0.253 | 0.153 |
| TS | 0.802 | 0.731 | 0.469 | 0.318 | 0.169 | 0.042 | 0.000 |
| FSS(15 km) | --- | 0.887 | 0.714 | 0.563 | 0.352 | 0.093 | 0.000 |
| FSS(25 km) | --- | 0.905 | 0.747 | 0.599 | 0.381 | 0.096 | 0.000 |
| FSS(45 km) | --- | 0.924 | 0.784 | 0.644 | 0.414 | 0.103 | 0.000 |
| FSS(75 km) | --- | 0.940 | 0.813 | 0.685 | 0.443 | 0.111 | 0.000 |
| FSS(120 km) | --- | 0.952 | 0.840 | 0.723 | 0.474 | 0.120 | 0.000 |




**Table 14.** PAS, TS and FSS scores of GRAPES 24-hour accumulated precipitation from 00:00 UTC on 19 July to

 00:00 UTC on 20 July 2021.

|  | clear/rainy | ≥0.1 mm | ≥10 mm | ≥25 mm | ≥50 mm | ≥100 mm | ≥250 mm |
|---|---|---|---|---|---|---|---|
| PAS | 0.665 | 0.549 | 0.396 | 0.414 | 0.494 | 0.573 | 0.338 |
| TS | 0.804 | 0.735 | 0.422 | 0.358 | 0.312 | 0.178 | 0.000 |
| FSS(15 km) | --- | 0.884 | 0.689 | 0.629 | 0.576 | 0.365 | 0.000 |
| FSS(25 km) | --- | 0.901 | 0.742 | 0.688 | 0.633 | 0.400 | 0.000 |
| FSS(45 km) | --- | 0.922 | 0.809 | 0.759 | 0.704 | 0.432 | 0.000 |
| FSS(75 km) | --- | 0.939 | 0.865 | 0.817 | 0.758 | 0.457 | 0.000 |
| FSS(120 km) | --- | 0.950 | 0.907 | 0.862 | 0.786 | 0.494 | 0.000 |


**Table 15.** PAS, TS and FSS scores of GRAPES 24-hour accumulated precipitation from 00:00 UTC on 20 July to 00:00 UTC on 21 July 2021.

|  | clear/rainy | ≥0.1 mm | ≥10 mm | ≥25 mm | ≥50 mm | ≥100 mm | ≥250 mm |
|---|---|---|---|---|---|---|---|
| PAS | 0.669 | 0.580 | 0.438 | 0.451 | 0.504 | 0.581 | 0.637 |
| TS | 0.801 | 0.746 | 0.544 | 0.438 | 0.431 | 0.451 | 0.045 |
| FSS(15 km) | --- | 0.891 | 0.774 | 0.693 | 0.683 | 0.687 | 0.127 |
| FSS(25 km) | --- | 0.909 | 0.808 | 0.737 | 0.727 | 0.721 | 0.167 |
| FSS(45 km) | --- | 0.930 | 0.850 | 0.793 | 0.787 | 0.767 | 0.218 |
| FSS(75 km) | --- | 0.947 | 0.884 | 0.843 | 0.847 | 0.818 | 0.233 |
| FSS(120 km) | --- | 0.960 | 0.913 | 0.885 | 0.897 | 0.864 | 0.238 |


**Table 16.** PAS, TS and FSS scores of GRAPES 24-hour accumulated precipitation from 00:00 UTC on 21 July to 00:00 UTC on 22 July 2021.

|  | clear/rainy | ≥0.1 mm | ≥10 mm | ≥25 mm | ≥50 mm | ≥100 mm | ≥250 mm |
|---|---|---|---|---|---|---|---|
| PAS | 0.694 | 0.566 | 0.407 | 0.425 | 0.462 | 0.492 | 0.528 |
| TS | 0.796 | 0.710 | 0.559 | 0.501 | 0.410 | 0.284 | 0.044 |
| FSS(15 km) | --- | 0.875 | 0.799 | 0.752 | 0.667 | 0.508 | 0.092 |
| FSS(25 km) | --- | 0.897 | 0.842 | 0.793 | 0.713 | 0.548 | 0.102 |
| FSS(45 km) | --- | 0.924 | 0.889 | 0.842 | 0.772 | 0.613 | 0.137 |
| FSS(75 km) | --- | 0.945 | 0.925 | 0.883 | 0.823 | 0.690 | 0.192 |
| FSS(120 km) | --- | 0.960 | 0.949 | 0.911 | 0.858 | 0.757 | 0.257 |




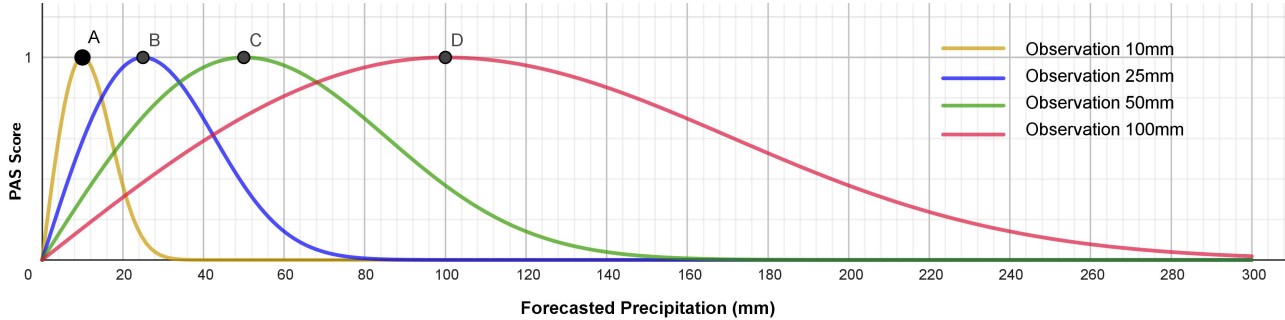


**Figure 1.** Schematic diagram of the precipitation forecast accuracy score (PAS) curves when the observed
precipitation amounts are 10, 25, 50 and 100 mm.


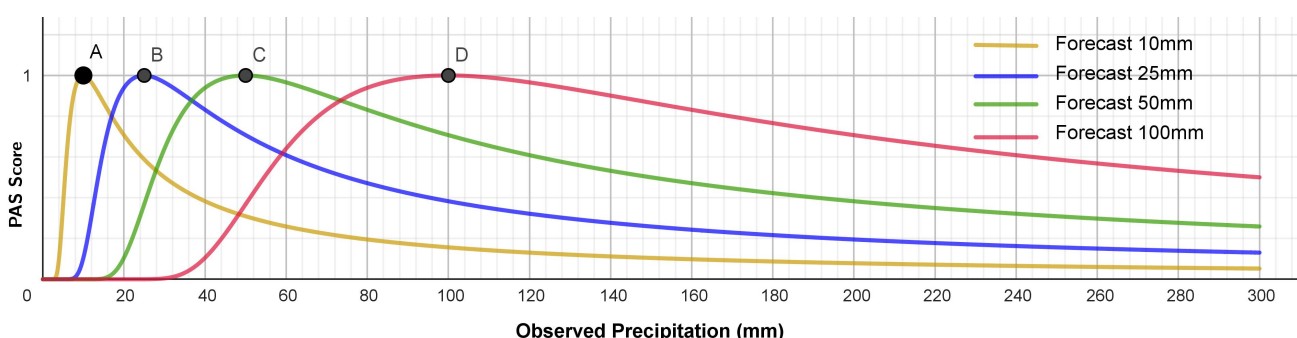


**Figure 2.** PAS curves corresponding to different forecasted precipitation amounts ($u$ = 10, 25, 50 and 100 mm).




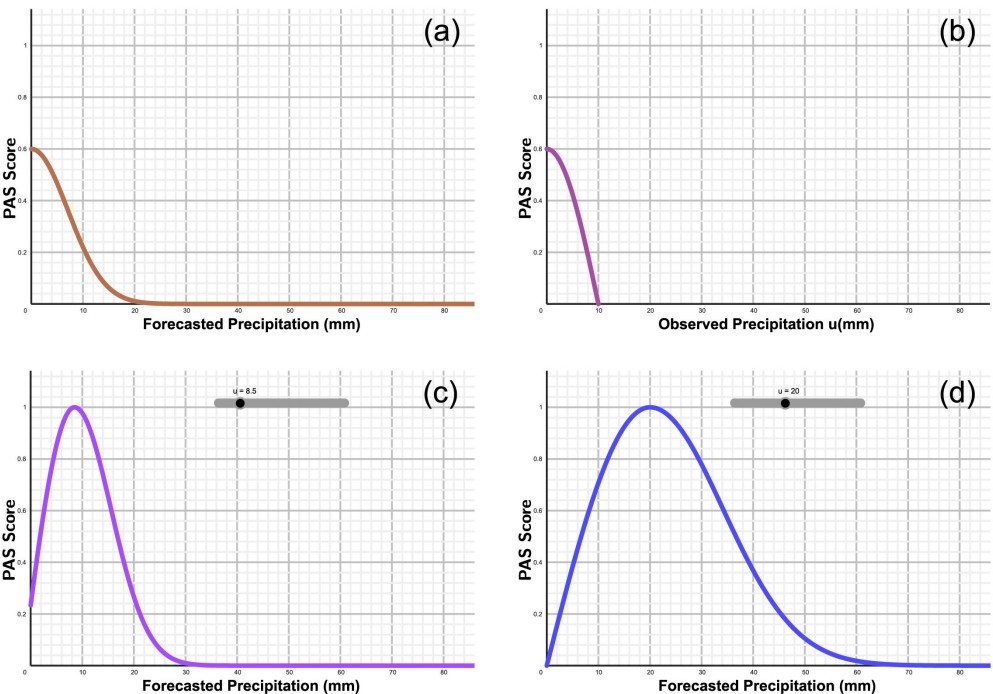


**Figure 3.** PAS curves of precipitation forecasts when (a) the observed precipitation u = 0 mm and the forecasted

precipitation x > 0 mm, (b) the observed precipitation 0 < u < 10 mm and the forecasted precipitation x = 0 mm

(the horizontal coordinate denotes the observed precipitation u), (c) the observed precipitation 0 < u < 10 mm and

the forecasted precipitation x > 0 mm, and (d) the observed precipitation u ≥ 10 mm.

866

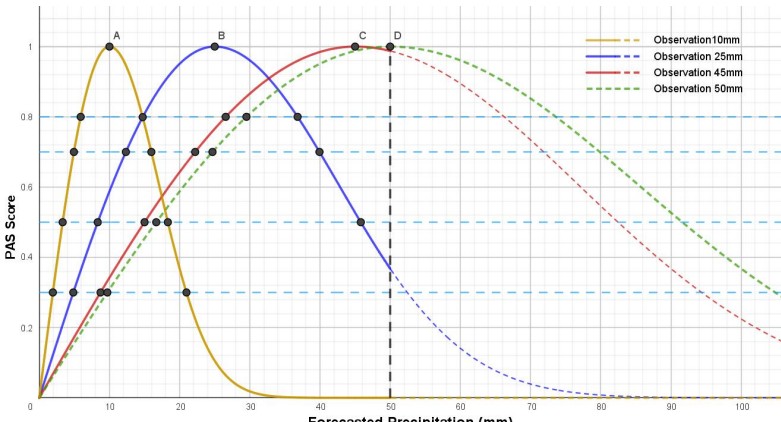

867

**Figure 4.** PAS curves of the forecasts under general precipitation conditions ($u$ = 10, 25 and 45 mm). The solid line

part of the curve in the figure is involved in the comparison, the dashed line part is not involved in the comparison,

10 mm observed precipitation is represented by the orange line, 25 mm observed precipitation is represented by the

blue line, 45 mm observed precipitation is represented by the red line, and 50 mm observed precipitation is

represented by the green line.

873

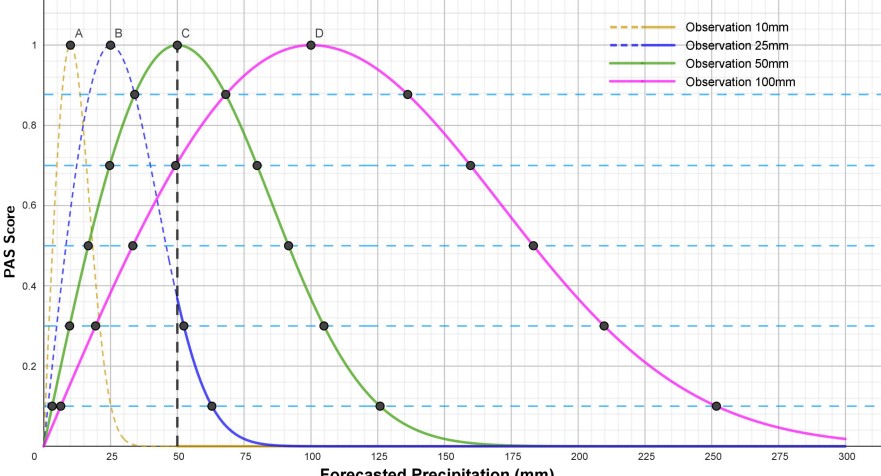

874

**Figure 5.** Same as Fig. 4, but for precipitation at the level of torrential rain and above ($u$ = 25, 50 and 100 mm). The solid line part of the curve in the figure is involved in the comparison, the dashed line part is not involved in the comparison, 10 mm observed precipitation is represented by the orange line, 25 mm observed precipitation is represented by the blue line, 50 mm observed precipitation is represented by the green line, and 100 mm observed precipitation is represented by the red line.


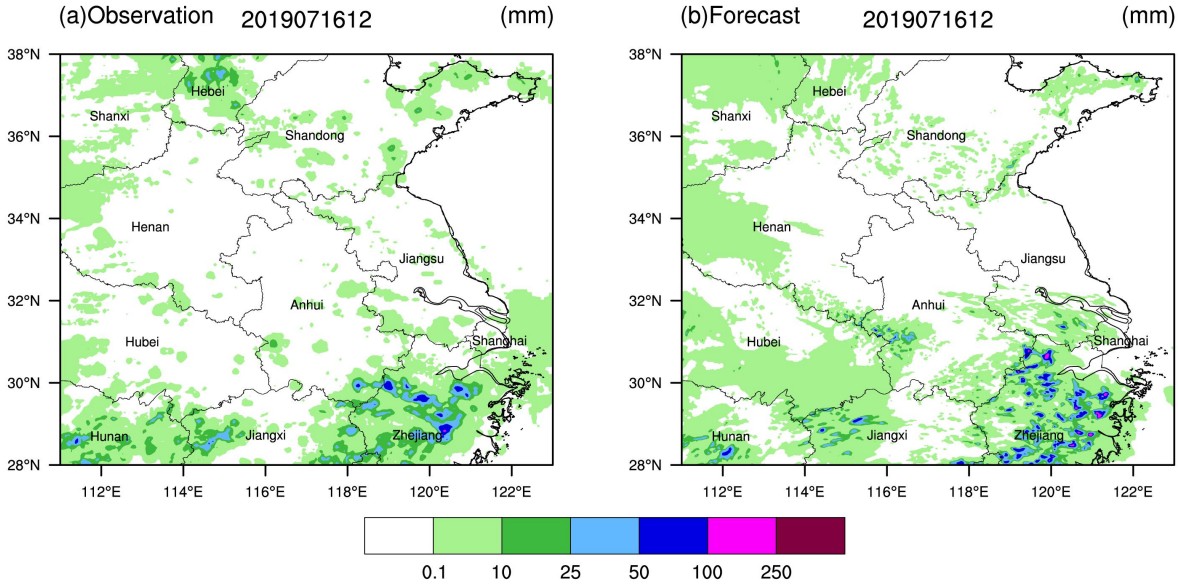


**Figure 6**. Accumulated precipitation (a) observed and (b) forecasted from 00:00 to 12:00 UTC on 16 July 2019.


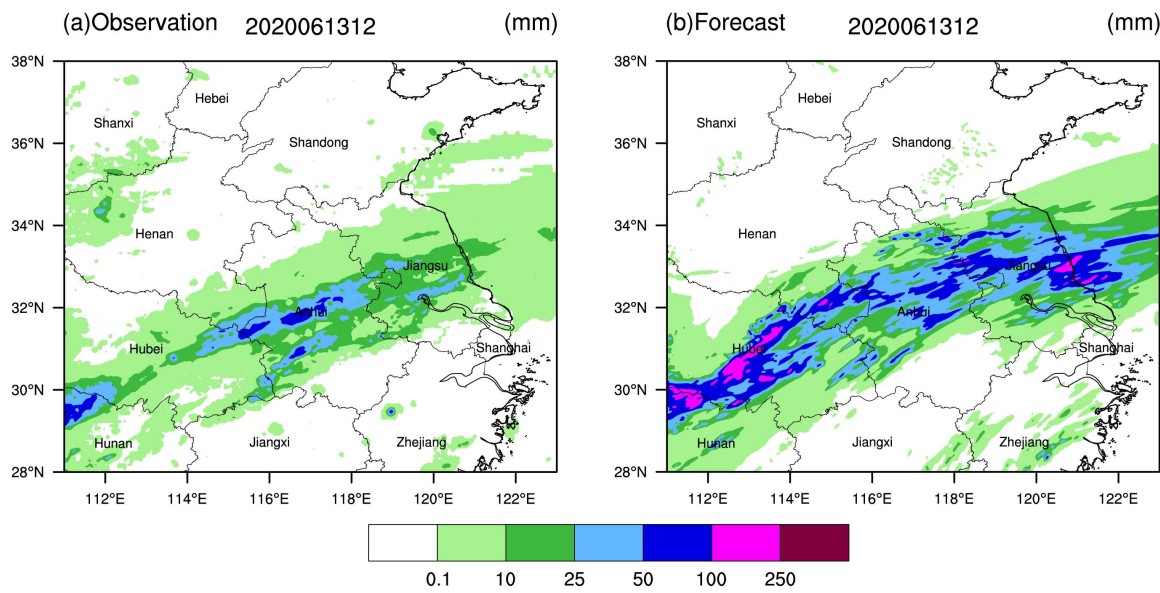


**Figure 7**. Accumulated precipitation (a) observed and (b) forecasted from 00:00 to 12:00 UTC on 13 June 2020.

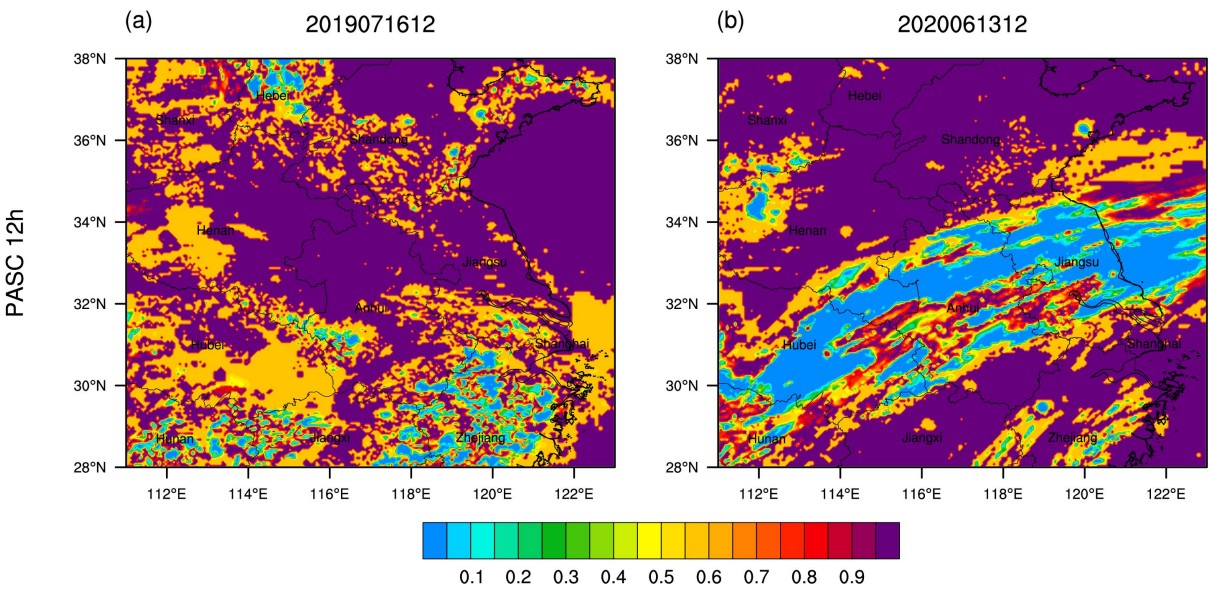


**Figure 8.** Distributions of the PAS clear/rainy forecast accuracy score (PASC) of 12-hour accumulated precipitation
for (a) Case 1 from 00:00 to 12:00 UTC on 16 July 2019 and (b) Case 2 from 00:00 to 12:00 UTC on 13 June 2020.

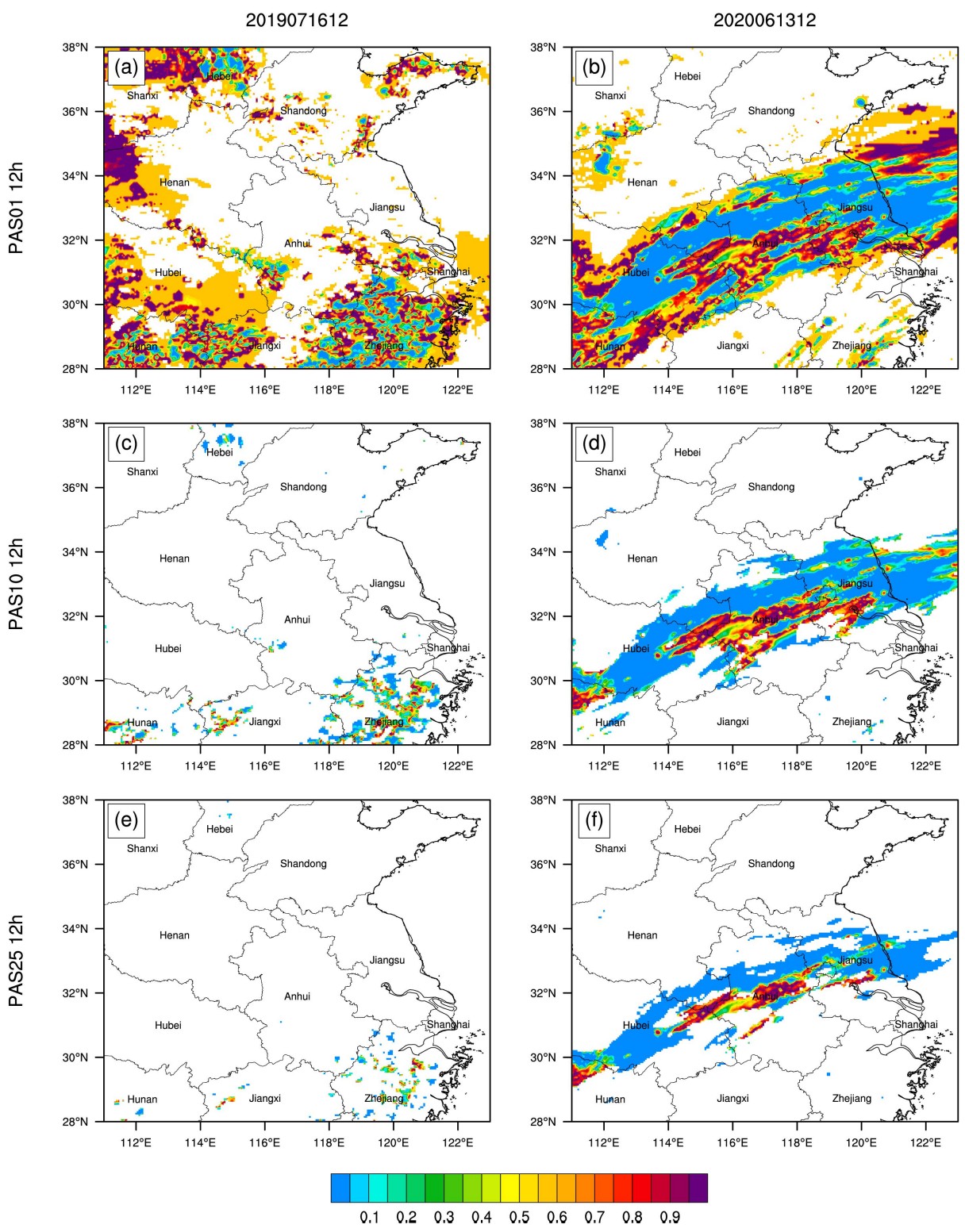


**Figure 9.** Distributions of PAS of 12-hour accumulated precipitation, ≥ 0.1 mm for (a) Case 1 from 00:00 to 12:00

UTC on 16 July 2019 and (b) Case 2 from 00:00 to 12:00 UTC on 13 June 2020, ≥10 mm for (c) Case 1 and (d)

Case 2, and ≥ 25 mm for (e) Case 1 and (f) Case 2.


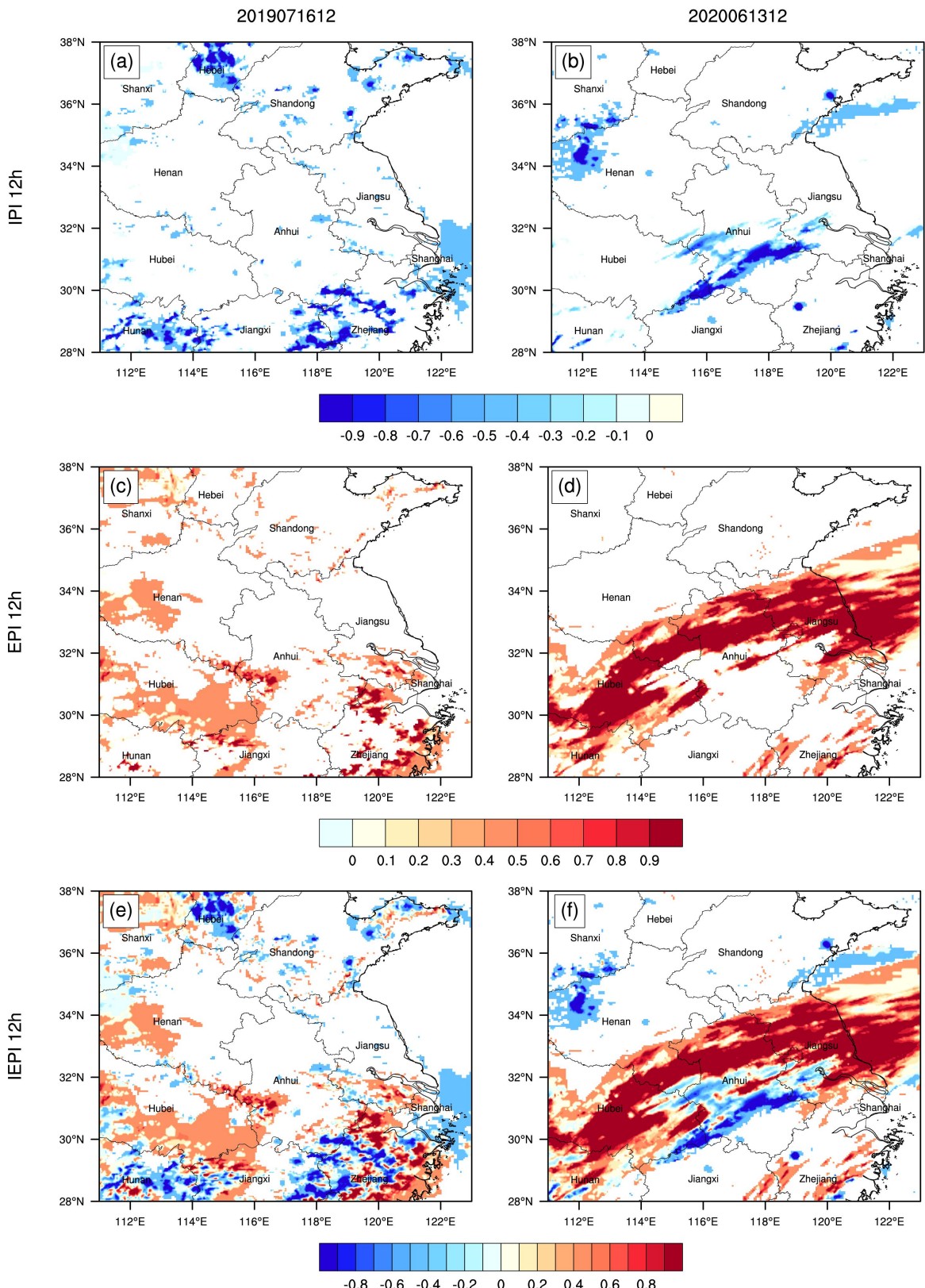


**Figure 10.** Distributions of IPI of 12-hour accumulated precipitation for (a) Case 1 from 00:00 to 12:00 UTC on

16 July 2019, and (b) Case 2 from 00:00 to 12:00 UTC on 13 June 2020, EPI for (c) Case 1 and (d) Case 2, and

IEPI for (e) Case 1 and (f) Case 2.

900

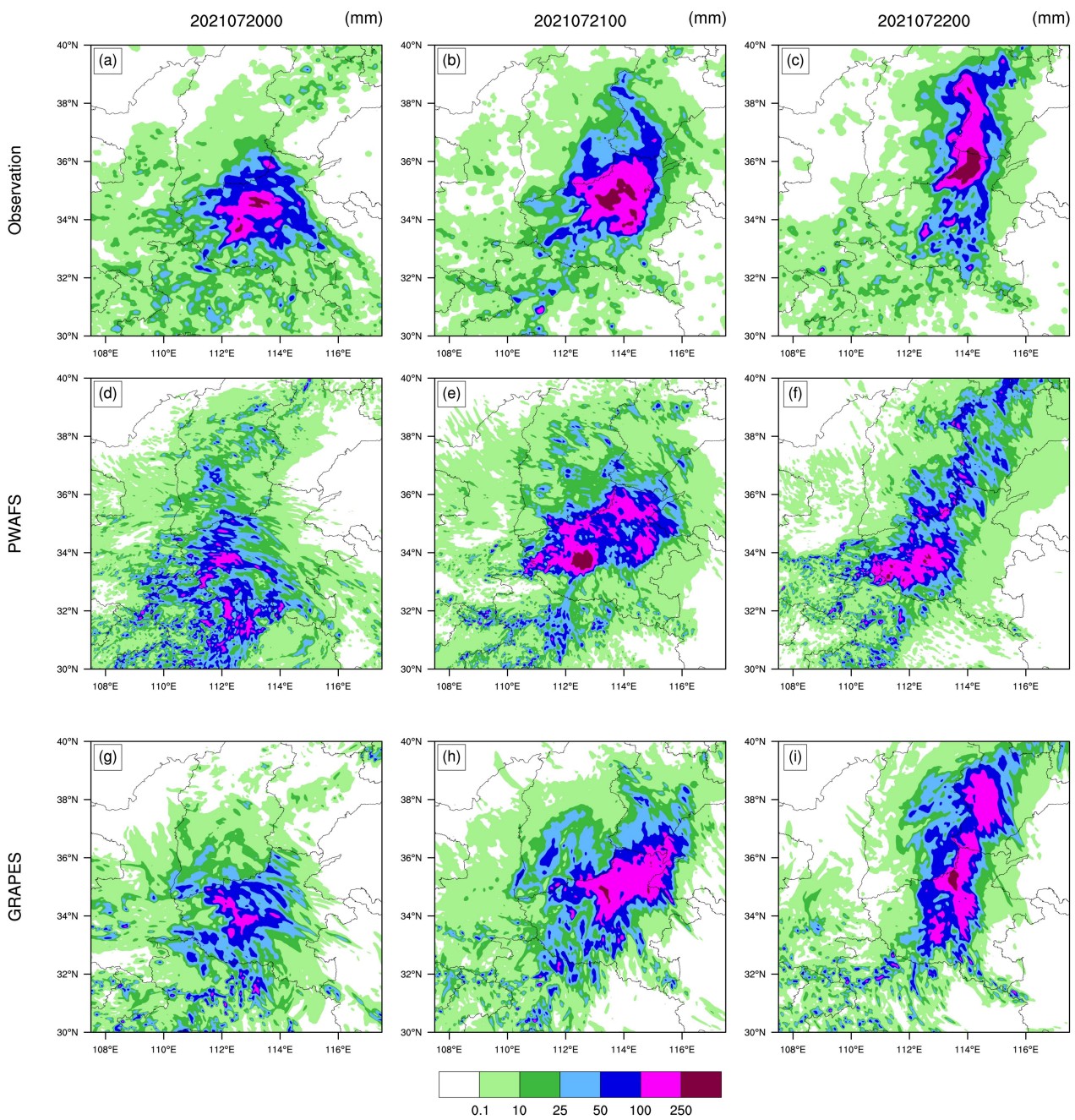

901

**Figure 11**. Distribution of observed and forecasted 24-hour accumulated precipitation. (a) Observation, (d) PWAFS,

(g) GRAPES from 00:00 UTC on 19 July to 00:00 UTC on 20 July 2021; (b) Observation, (e) PWAFS, (h)

GRAPES from 00:00 UTC on 20 July to 00:00 UTC on 21 July 2021; (c) Observation, (f) PWAFS, (i) GRAPES

from 00:00 UTC on 21 July to 00:00 UTC on 22 July 2021.

906