# Peer review of "A General Comprehensive Evaluation Method for Cross-Scale Precipitation Forecasts"

_EGUsphere, 2023_

## Author Comment (AC1)

Dear editor of GMD:

We are now resubmitting the revised manuscript "A General Comprehensive Evaluation Method for Cross-Scale Precipitation Forecasts" [egusphere-2023-2613] to Geoscientific Model Development. We thank the reviewers and editors for their insightful and constructive suggestions, which helped us to improve this manuscript greatly. Our point-by-point responses to the reviewers' comments are provided below.

Sincerely,

Anning Huang

School of Atmospheric Sciences, Nanjing University, China

**Referee #1**

**General comments**

In this article, a new evaluation method named GCEM is introduced for assessing the accuracy of precipitation forecast. This method employs a scoring function that produces more continuous scores compared to the traditional threat score (TS). The authors present GCEM as a general method suitable for cross-scale precipitation forecasts, including both general and extreme precipitation events. The advantages of GCEM, in terms of providing more continuous scores and more spatial information, are demonstrated by analyzing the distribution of several relevant indices in two typical cases. While GCEM offers an alternative to TS with a simple and concise formula, the authors should provide detailed insights into understanding the practicality and limitations of the new method, especially concerning extreme events. Furthermore, additional comparisons with other advanced methods are necessary to substantiate the effectiveness of GCEM. Overall, the manuscript is well written and makes a valuable contribution to the field of precipitation forecast evaluation.

Therefore, I would recommend the article for publication after minor revisions according to the specific comments below.

**Response:** We warmly appreciate the expert for taking the precious time to help us greatly improve the manuscript. We appreciate the reviewer's positive comments on the manuscript. We have revised the manuscript following the constructive suggestions, and the revised details are listed as follows:

**Specific comments**

1. P1 L17: Short summary: "This method does not utilize the traditional contingency table-based classification verification, can replace the threat score (TS), equitable threat score (ETS) and other skill score methods ...". What do the other skill score methods refer to? How does GCEM compare with other modern forecasting evaluation methods? Spatial distribution of the other advanced indices or overall scores could be beneficial to demonstrate the advantages of GCEM.

**Response:** Thank the expert for this constructive suggestion. "Other skill score methods" also refer to scoring indices based on categorical contingency tables, such as the Peirce skill score (PSS), Heidke skill score (HSS), probability of detection (POD), frequency bias (BIAS), accuracy (ACC), false alarm ratio (FAR), missing ratio (MR), and probability of false detection (POFD).

Since traditional scores often do not reflect model performance improvements, modern forecast verification is based mainly on spatial verification methods to compensate for the shortcomings some traditional methods. The literature review by Gilleland et al. (2009a) defines four main categories of spatial verification methods: neighbourhood, scale separation, features based, and field deformation. The neighbourhood approaches involve selecting different space-time neighbourhood to perform validation statistics on the elemental field, the scale separation approaches examine forecast performance by separating features at each scale, and the feature-based approaches identify features

in the forecast and observation fields and then evaluate different attributes (e.g., location, magnitude, and intensity) associated with each pair of forecast-observation features. The field deformation approaches analyze the spatial manipulation of the forecast field to make it look as much like the observation field as possible. GCEM is based on point-to-point scoring statistics, without a radius of influence, no isolation of features at each scale, and no definition of objects in the forecast and observation to analyze the similarity of the objects or to fit the forecast objects through deformation operations. However, the GCEM still has spatial attributes that can discriminate the spatial forecast characteristics (e.g., insufficient or excessive forecasting scenarios) for different categories of precipitation. In addition, these modern spatial verification methods can perform more comprehensive analysis in specific individual cases, but appear to be less able to provide direct overall scoring results than traditional scoring methods in the statistics of long time series, whereas the GCEM can carry out statistical verification of long time series and produce overall scoring results.

We have added the statements at P5 L92-96: "*In addition to the TS and ETS, the methods of traditional contingency table-based classification verification include Peirce skill score (PSS) (Peirce, 1884; Hanssen and Kuipers, 1965; Murphy and Daan, 1985; Flueck, 1987), Heidke skill score (HSS) (Doolittle, 1885; Doolittle, 1888; Heidke, 1926), probability of detection (POD), frequency bias (BIAS), accuracy (ACC), false alarm ratio (FAR), missing ratio (MR), probability of false detection (POFD), etc.*"

P22 L466-477: "*Modern forecast verification is based mainly on spatial verification methods to compensate for the shortcomings of traditional methods. The literature review of Gilleland et al. (2009a) defines four main categories of methods: neighbourhood, scale separation, features based,*

*and field deformation. These methods can analyze more comprehensively in specific individual cases, but seem to be less able to provide direct overall scoring results than traditional scoring methods in the statistics of long time series. GCEM is based on point-to-point scoring statistics, without a radius of influence, no isolation of features at each scale, and no definition of objects in the forecast and observation to analyze the similarity of the objects or to fit the forecast objects through deformation operations. However, the GCEM still has spatial attributes that can discriminate spatial forecast characteristics (e.g., insufficient or excessive forecasting scenarios) for different categories of precipitation, and the GCEM can carry out statistical verification of long time series and produce overall scoring results.*"

2. P14 L304, L306, L309, L312: I suggest that the authors provide a rationale for choosing a coefficient of 0.6 under the given condition. Is PAS equal to 1 or 0.6 when u=0 and x=0? Further clarification on the extent of overall PAS value's sensitivity to the coefficient would help understand the robustness of GCEM and its applicability to light rain.

**Response:** Thank the expert for alerting us to this. The questions you raised are of great practical significance. Due to the length of the article, the reason why the selection coefficient is 0.6 is not carefully indicated. The determination of the coefficient is based on the case of u=0 mm and x=0 mm, and its score is 0.6 times the normal score. In actual observations, the minimum observed value of precipitation is 0.1 mm, so precipitation <0.1 mm is equivalent to 0.0 mm (i.e., no precipitation).

In the first case, when the observation has no precipitation (u=0 mm) and the prediction has precipitation (x $\neq$ 0 mm), it is a failure of the forecast for transitional weather. For example, the situation with the observation u=0 mm and forecast x=0.1 mm is qualitatively different from the

situation with the observation u= 0.1 mm and forecast x= 0.2 mm. The former situation is the failure of the "≥0.1 mm precipitation forecast", the latter is correct. Therefore, for the case in which the observation has no precipitation (u=0 mm) and the forecast has precipitation (x≥0.1 mm), we set a coefficient of 0.6 as an appropriate penalty. Thus, for the case in which the observation has no precipitation (u=0 mm) and the forecast precipitation is 0.1 mm (x=0.1 mm), PAS=0.6 according to Eq. (6) $PAS = 0.6PAS_{|u\to0}$, implying that the forecast effect has just reached the standard, like the ACC reaching 0.6 indicates that the model forecast effect is available (Zhao et al., 2018). For the other cases in which the observation has no precipitation (u=0 mm) and the forecast precipitation is greater than 0.1 mm (x>0.1 mm), PAS<0.6.

In the second case, when the observation has precipitation (0<u<10 mm) and the prediction has no precipitation (x=0 mm), PAS≤0.6 indicates the failure of the transitional weather forecast. The best case is that the observed precipitation is 0.1 mm (u=0.1 mm) and the forecast has no precipitation (x=0 mm). According to Eq. (7), PAS=0.6 indicates that the forecast effect has just reached the standard. For the other cases in which the observed precipitation is greater than 0.1 mm (u>0.1 mm) and the forecast has no precipitation (x=0 mm), PAS<0.6 according to Eq. (7).

For the typical cases in the manuscript, the adjustment coefficients of the PAS are set to 1.0 (PAS1.0) and 0.5 (PAS0.5). A comparison of PAS1.0 and PAS0.5 with PAS0.6 (the coefficient is set to 0.6), which has a coefficient of 0.6, shows that the scores for precipitation ≥0.1 mm are more sensitive to the adjustment coefficients, while the scores for precipitation ≥10 mm are almost unaffected. According to the scoring results (Tables 1 and 2), whether the adjustment coefficients are 0.5, 0.6, or 1.0, they will not affect the conclusions in the manuscript, nor will they affect the applicability to light rain. In summary, the adjustment coefficient is set to 0.6, which can maintain

the robustness of the GCEM and its applicability to light rain.

**Table 1.** PAS and TS of 12 h accumulated precipitation from 00:00 to 12:00 UTC on 16 July 2019.

|  | Clear/rainy | ≥0.1 mm | ≥10 mm | ≥25 mm | ≥50 mm |
|---|---|---|---|---|---|
| PAS0.5 | 0.779 | 0.559 | 0.255 | 0.200 | 0.104 |
| PAS0.6 | 0.808 | 0.617 | 0.256 | 0.200 | 0.104 |
| PAS1.0 | 0.923 | 0.846 | 0.260 | 0.200 | 0.104 |
| TS | 0.690 | 0.381 | 0.194 | 0.076 | 0.006 |

**Table 2.** Same as Table 8, but from 00:00 to 12:00 UTC on 13 June 2020.

|  | Clear/rainy | ≥0.1 mm | ≥10 mm | ≥25 mm | ≥50 mm |
|---|---|---|---|---|---|
| PAS0.5 | 0.719 | 0.425 | 0.227 | 0.185 | 0.116 |
| PAS0.6 | 0.734 | 0.457 | 0.228 | 0.185 | 0.116 |
| PAS1.0 | 0.797 | 0.584 | 0.230 | 0.185 | 0.116 |
| TS | 0.816 | 0.625 | 0.338 | 0.149 | 0.036 |

The modifications in the revised manuscript are as follows:

P15 L318-321: "*The coefficient was set to 0.6. According to Eqs. (6-7), when the situation is the observation u=0 mm and forecast x=0.1 mm or the observation u=0.1 mm and forecast x=0 mm, PAS=0.6, suggesting that the forecast effect has just reached the standard, just like when the ACC reaches 0.6, which indicates that the model forecast effect is available (Zhao and Zhang, 2018).*"

References:

Zhao, B., and Zhang, B.: Application of neighborhood spatial verification method on precipitation evaluation[J], Torrential Rain and Disasters,37(1):1-7, DOI: 10.3969/j.issn.1004-9045.2018.01.001, 2018.

3. P18 L371: The comparison of the two typical cases is clear but not enough to generalize the effectiveness of the general method across different types of precipitation events, geographical areas, or other meteorological conditions. I suggest examining the method in a broader selection of cases covering a range of conditions, especially for extreme precipitation (e.g., extreme rainfall event over

Henan in July 2021). Meanwhile, analyzing results from different forecast models, if available, could be beneficial. This approach would allow us to discuss how GCEM can help identify specific problems in each model that TS cannot.

**Response:** The expert has proposed studies on the adaptability of the GCEM using different cases, especially its ability to evaluate extreme precipitation events. We thank you for this nice suggestion. According to the suggestion, we have added the PAS, TS and FSS methods to evaluate the forecast of extreme precipitation events in Henan, China, in July 2021 with different high-score models to understand and analyze the differences among the different scoring methods, and to further grasp the characteristics of the GCEM.

The process of the Henan extreme precipitation event is divided into three periods: (1) 00:00 UTC on 19 July - 00:00 UTC on 20 July 2021. (2) 00:00 UTC on 20 July - 00:00 UTC on 21 July 2021. (3) 00:00 UTC on 21 July - 00:00 UTC on 22 July 2021. The precipitation predicated by two regional models (PWAFS and GRAPES) with a horizontal resolution of 3 km is used for evaluation, and the precipitation is the accumulated precipitation of 24 hours in the forecast 12-36 hours. The evaluation results are as follows:

[revised manuscript text omitted]

**P24-29 L519-615: Section 4.2 has been added to the manuscript to illustrate this example.**

References:

Chen, F., Chen, J., Wei, Q., Li, J., Liu, C., Yang, D., Zhao, B., and Zhang, Z.: A new verification method for heavy rainfall forecast based on predictability II: Verification method and test[J], Acta Meteorologica Sinica, 77(1): 28-42. doi: 10.11676/qxxb2019.003, 2019.

4. P20 L430: "…, which differs from subjective judgement" How do the authors define the subjective judgement (whether it be from forecasters, the public, or model developers)? Why is TS different from subjective judgment, perhaps due to the double penalty issue? The authors are encouraged to further explain how GCEM addresses and overcomes the limitations of TS in these cases. To enhance understanding, a comparison of GCEM with other advanced methods that avoid the double penalty issues would be valuable.

**Response:** Thank the expert for the suggestions and reminders. The subjective evaluations of the two typical cases mentioned in the manuscript come from model developers and the consulted public. Because the differences between the two typical cases are relatively direct, no subjective evaluation conclusion is drawn through statistical investigation. The TS is different from subjective judgement and may be related to the double penalty.

The expert suggested that we further understand the characteristics and advantages of the

GCEM method by comparing it with other scoring methods that avoid the double penalty. Therefore, the previous question was utilized to analyze the effectiveness of the GCEM in verifying extreme precipitation events. The FSS belongs to the neighbourhood category of spatial verification methods and is an advanced evaluation method widely used in recent years. It can still yield valuable scores when the model prediction intensity is spatially biased and can also represent the scale information of forecasting skills. Therefore, in this case, the FSS scoring method was added for comparative experiments. For FSS verification, 15 km, 25 km, 45 km, 75 km and 120 km are used as the neighbourhood distances.

In this example, 52 questionnaires were completed by 32 researchers and 20 forecasters. The names of the PWAFS and GRAPES models used for comparison were omitted and replaced with Model 1 and Model 2, respectively.

The three periods [(1) 00:00 UTC on 19 July - 00:00 UTC on 20 July 2021. (2) 00:00 UTC on 20 July - 00:00 UTC on 21 July 2021. (3) 00:00 UTC on 21 July - 00:00 UTC on 22 July 2021] of the extreme precipitation event in Henan Province from 19-22 July 2021 are referred to as period A, period B, and period C, respectively. The survey results are as follows: 52 people believe that Model 2 (GRAPES) has a good forecast effect for periods A and C, 19 people believe that Model 1 (PWAFS) has a good forecast effect for period B, and 33 people believe that Model 2 (GRAPES) has a good forecast effect for period B. Fifty-two people think that Model 2 (GRAPES) is good in general.

For objective scores, the PAS scores are consistent with the subjective scores at all levels. The TS and FSS scores are mostly consistent with the subjective evaluations but with some scores for clear/rain, $\geq 0.1$ mm, $\geq 25$ mm and $\geq 50$ mm deviating from subjective evaluations. Since subjective evaluations are less detailed than objective evaluations, these biases are normal. A comparison

between the PAS and FSS reveals that in small-scale precipitation assessment, the FSS scores of the precipitation amounts $\geq 0.1$ mm and $\geq 10$ mm are always high due to the expansion of the field. The FSS avoids the double penalty and increases the redundancy, which makes it difficult to distinguish the prediction effect of precipitation $\geq 0.1$ mm. However, the FSS scores of intense precipitation (especially $\geq 250$ mm) is much lower, and it is at the unskilled end, making it difficult to effectively distinguish the difference in the model prediction effect.

Due to the use of continuous function scoring methods, the PAS can effectively evaluate the forecast performance of the model for both precipitation $\geq 0.1$ mm and intense precipitation ( $\geq 100$ mm), so it has certain advantages compared with the currently widely used advanced scoring methods.

According to the suggestions of the expert, we have added a typical case of extreme precipitation in Henan, China, to illustrate that the PAS has advantages for both the TS and FSS. Of course, the comprehensive use of multiple scoring methods is conducive to a comprehensive understanding of the performance indicators of the model.

**P24-29 L519-615: Section 4.2 has been added to the manuscript to illustrate this example.**

**Technical corrections**

1. P3 L52: Short duration heavy rainfall often leads to …

**Response:** Done.

2. P10 L214: … its core scoring function is as follows.

**Response:** Done.

3. P14 L301: The sentence is confusing.

**Response:** Thank you for alerting us to this. This sentence is not appropriate and has been deleted.

4. P15 L313 writes "when 0<u<10" while L315 Eq. 9 writes "0<=u<10". Please check whether it is correct.

**Response:** It has been revised in the manuscript as follows:

P15 L324: "$PAS = \begin{cases} sin\left(\frac{\pi}{2} \cdot \frac{x-u+10}{10}\right), & 0 < x < u, \ 0 < u < 10 \\ e^{-\left(\frac{x-u}{10}\right)^2}, & u \leq x, \quad 0 < u < 10 \end{cases}$   (8)"

5.P35 Fig. 4: The figure title can be more concrete by indicating representations of different styles (solid/dashed) of lines.

**Response:** The explanation for Figure 4 has been added.

6.P36 Fig. 5: The same issue as in Fig. 4.

**Response:** The explanation for Figure 5 has been added.

**Referee #2**

Comments on the manuscript entitled "A General Comprehensive Evaluation Method for Cross-Scale Precipitation Forecasts" by Zhang et al. submitted to GMD.

**General comments:**

The authors propose a novel precipitation evaluation method, the Precipitation Accuracy Score (PAS), diverging from traditional Threat Score (TS). PAS utilizes continuous functions for more precise precipitation forecast accuracy measurement, constituting a notable advancement in the evaluation method of precipitation forecasts. The study falls in the scope of GMD. However, the manuscript is hard to follow due to vague expressions and English writing. A substantial revision is imperative to enhance clarity and facilitate comprehension before the manuscript can be deemed

suitable for publication in GMD. Specific comments are as follows:

**Response:** We sincerely thank you for greatly helping us to improve the manuscript. We are encouraged by your positive comments on the manuscript. We have revised the manuscript following the beneficial comments and the details are shown as follows:

**Specific comments:**

1. L63: Clarify why conventional scoring methods fail to reflect model performance improvements. Specify the methods encompassed within the "traditional scoring methods" category.

**Response:** Thank you for this comment. With the advancement of computing power, the grid spacing of numerical weather forecasts has become increasingly refined. Although the simulated precipitation distribution has a more realistic spatial structure and can provide valuable guidance to forecasters on convective evolution, small errors in the position or timing of small convective features can lead to false alarms and missed events in traditional scoring methods, resulting in a dual penalty problem and a lower final forecast score, which cannot reflect model performance improvements (Ahijevych et al., 2009; Chen et al., 2021).

The specific traditional scoring methods refer to the threat score (TS), equitable threat score (ETS), Peirce skill score (PSS), Heidke skill score (HSS), probability of detection (POD), frequency bias (BIAS), accuracy (ACC), false alarm ratio (FAR), missing ratio (MR), probability of false detection (POFD), etc., which are based on categorical contingency tables.

The modifications at P5 L92-96 in the manuscript are as follows:

"*In addition to the TS and ETS, the methods of traditional contingency table-based classification verification include Peirce skill score (PSS) (Peirce, 1884; Hanssen and Kuipers, 1965; Murphy and Daan, 1985; Flueck, 1987), Heidke skill score (HSS) (Doolittle, 1885; Doolittle, 1888;*

*Heidke, 1926), probability of detection (POD), frequency bias (BIAS), accuracy (ACC), false alarm*

*ratio (FAR), missing ratio (MR), probability of false detection (POFD), etc.*"


**Response:** Thank the expert for the suggestions. Case "x=u" cannot be merged into Case "0≤x≤u", as u=0 can not be divisor. Case "x=u" can be merged into Case "0<u≤x", which has been

modified as follows:

P12 L245: "$IEPI = \begin{cases} sin\left(\frac{\pi}{2}\cdot\frac{x}{u}\right) - 1, \ 0 \le x < u \\ 1 - e^{-\left(\frac{x-u}{u}\right)^2}, \quad 0 < u \le x \end{cases}$ (4)"

We note that the condition "x=u" in the original Eq. (4) should be changed to "0<x=u" because "x=u=0" is a correctly forecasted non-precipitation event and is not within the calculation range of IEPI.

Regarding the second question, if we look at the exponential distribution of a two-dimensional plane, the IPS and EPS (IPI and EPI in the revised manuscript) are indeed included in IEPS (IEPI in the revised manuscript). However, if the overall bias is calculated, the IEPI can only reflect the overall forecast deviations but the forecast deviation information reflected after separation is more abundant.

Eq. (4) is divided into three classifications: "0≤x<u" for insufficient precipitation forecasting (underestimation), "0<x=u" for perfect forecasting (no deviation), and "0<u<x" for excessive precipitation forecasting (overestimation).

Let p be the number of grid points where the forecast is less than the observed value, q be the number of grid points where the forecast is greater than the observed value, and w be the number of precipitation grid points where the forecast is equal to the observed value, n=p+q+w. IPI $_{(i)}$, EPI $_{(i)}$, and IEPI$_{(i)}$ represent the insufficient, excessive, and bias indicators of single point i, respectively (suppose it exists but not at the same time).

Then,

$$IPI = \frac{\sum_{i=1}^{p} IPI_{(i)}}{p} \tag{1}$$

$$EPI = \frac{\sum_{i=1}^{q} EPI_{(i)}}{q} \tag{2}$$

$$IEPI = \frac{\sum_{i=1}^{n} IEPI_{(i)}}{n} = \frac{\sum_{i=1}^{p} IPI_{(i)} + \sum_{i=1}^{q} EPI_{(i)} + \sum_{i=1}^{w} IEPI_{(i)}}{n} \tag{3}$$

According to the IPI, EPI, and IEPI, these indices reflect different forecast bias information. The IPI is the sum of the IPIs of the grids with underestimated precipitation divided by the total number of grids with underestimated precipitation (p grids), and the EPI is the sum of the EPIs of the grids with overestimated precipitation divided by the total number of grids with overestimated precipitation (q grids). The IEPI is the sum of the IPIs and EPIs divided by the number of insufficient,

excessive, and perfect forecasts. In general, IPI + EPI ≠ IEPI.

For the typical example of Section 4.1, in Case 1, there are 10436 IPI points with xu and 1 point with x=u, so there are 24046 IEPI points. In Case 2, there are 6391 IPI points with xu, and 0 points with x=u, so there are 23472 IEPI points. As shown in Table 9, for Case 1, the absolute value of IEPI is 0.057, and the absolute values of IPI and EPI are not very different. For Case 2, the IEPI is 0.325, which is mainly caused by the large EPI value. Therefore, the IPI, EPI and IEPI reflect different forecast deviations, and the IPI and EPI are not redundant.

**Table 9.** The indices of insufficient precipitation forecast (IPI), excessive precipitation forecast (EPI) and insufficient and excessive precipitation forecast (IEPI) of 12-hour accumulated precipitation for two precipitation processes.

|        | IPI    | EPI   | IEPI  |
|--------|--------|-------|-------|
| Case 1 | -0.376 | 0.389 | 0.057 |
| Case 2 | -0.400 | 0.597 | 0.325 |

7. L246-249: Given the definition of PASN, the interval of x and u in Eq. (2-4) may change accordingly, e.g. replace "0<u≤x" with "0.1<u≤x"? PASN includes PAS|ux0.1 when u ≥ 0.1 or x ≥ 0.1. Please clearly define the intervals to distinguish PASN from PAS|ux0.1.

**Response:** Thank the expert for those nice suggestions. Eqs. (2-4) are a series of theoretical indicator formulas derived from Eq. (1); therefore, Eqs. (2-4) are referred to as the core calculation formulas for the IPI, EPI, and IEPI, respectively, and their interval definitions are suitable in this case. In practical applications, the optimized solution will be used (see Section 2.2) to calculate the IPI, EPI, and IEPI for u ≥ 0.1 mm or x ≥ 0.1 mm.

The relevant supplementary explanations have been added to section 2.1 of the manuscript as follows:

P12 L251-254: "*Eqs. (2-4) are a series of theoretical indicator formulas derived from Eq. (1),*

*therefore Eqs. (2-4) are referred to as the core calculation formulas for the IPI, EPI, and IEPI, respectively. In practical applications, the optimized solution will be used (see Section 2.2) to calculate the IPI, EPI, and IEPI for the situations of $u \geq 0.1$ mm or $x \geq 0.1$ mm."*

In addition, the original expression Eq. (5) is the overall PAS clear/rainy forecast accuracy score (PASC) of the verification domain. Here, we find that using this formula appears somewhat cumbersome and unclear. In actual observations, the minimum observed precipitation value is 0.1 mm, so precipitation < 0.1 mm is equivalent to 0.0 mm (i.e., no precipitation). Using the following expression ensures consistency between intervals, and the calculation results of the PASC will not change.

Modifications at P12 L257-261: *"*$$PASC = \begin{cases} 1 & 0 \leq u < 0.1 \text{ and } 0 \leq x < 0.1 \\ PAS_{|ux0.1} & u \geq 0.1 \text{ or } x \geq 0.1 \end{cases} \qquad (5)$$

*where PASC represents the PAS scoring value for clear/rainy forecasts. "$0 \leq u < 0.1$ and $0 \leq x < 0.1$" denotes the correctly forecasted non-precipitation event with PASC=1. $PAS_{|ux0.1}$ denotes the overall PAS for precipitation forecasts under specific conditions where the observed precipitation $u \geq 0.1$ mm or the forecasted precipitation $x \geq 0.1$ mm."*

8. L285-286: The authors mentioned in L285-286 that x represents observed precipitation and u stands for forecasted precipitation, which contradicts the information in L216-217. Should the authors confirm if there's an intention to interchange x and u to assess the symmetry of PAS in this context?

**Response:** Thank the expert for this nice suggestion. We usually treat x as the independent variable and treat other variables such as u as parameters. Therefore, in Eq. (1), the predicted precipitation is expressed using x, and evaluating the symmetry of the PAS naturally leads to the idea of exchanging

x and u. As mentioned by the expert, this contradicts the information in L216-217 (the original manuscript). Therefore, it is possible to declare the observed precipitation u as the independent variable without changing the sign of the variable and obtain the same result with moderate symmetry.

The modifications in the revised manuscript are as follows:

P14 L293-296: "*Moderate symmetry. In Eq. (1), let the observed precipitation is the independent variable $u$, and the forecasted precipitation is the parameter $x$. Similarly, for different magnitudes of forecasted precipitation (parameter $x$ = 10, 25, 50 and 100 mm) and observed precipitation (variable $u$) ranging from 0 to 300 mm, the corresponding scores are shown in Fig. 2.*"

9. L306: Does PAS=0.6PAS|u->0 or PAS=0.6PAS|x->0 in L306?

**Response:** Thank you for alerting us to this. There was an error, which was revised to:

P15 L310: "$PAS = 0.6PAS_{|x\to0}$ when $x$ = 0 mm, and $0 < u < 10$ mm;"

10. L380-381: For Fig. 6, kindly label the location of Hunan, Jiangxi, and Zhejiang Provinces or provide the longitude and latitude range.

**Response:** Thank you for this suggestion. Figure 6, as well as Figures 7, 8, 9 and 10, have been labeled with provincial administrative divisions.

11. L414-416: The forecasted precipitation was interpolated onto observed grid points. Therefore, the forecasted and observed data should share the same grid points. How to understand "forecasted data on the grid point nearest to the observed grid point"? Did the authors employ a neighbourhood

verification method? Please clarify.

**Response:** Thank you for alerting us to this. Yes, by interpolating the forecasted precipitation to the observed grid points, the forecasted and observed data share the same grid points when performing the precipitation verification.

The modifications in the revised manuscript are as follows:

P20 L431-433: "*Specifically, the forecasted data on the model grid nearest to the observed grid are used as the forecasted value at this observed grid.*"

The precipitation data in this example are grid data of longitude and latitude grids, with a spatial resolution of 0.05 ° × 0.05 °, while the forecasted precipitation data are 3 km equidistant grid point data. The forecasted precipitation data have a smaller grid distance than the observed data. Considering that precipitation verification is conducted on the grid points of observation data, as the forecasted data have a higher resolution than the observed data, the interpolation of forecasted data to the observed grid points adopts the nearest neighbour method. The WMO recommends the method of proximity point matching in the QPF verification technical rules for business models (WWRP/WGNE Joint Working Group on Verification, 2008). This method was also used in the precipitation performance verification of high-resolution models by the China Meteorological Administration. The sample matching method in the "Assessment Report on the Authenticity of Precipitation Grid Real Situation Products" states that "the observed data from the inspection source is taken as the 'true value', and the interpolation method of natural proximity is used to select the grid real situation value closest to the station and interpolate it to the observation point for matching." In high-resolution situations, this method is more reasonable and can avoid significant errors caused by interpolation processes on variables with poor continuity such as precipitation (Xu et al., 2017).

In this example, the precipitation evaluation is a point-to-point verification and does not employ a neighbourhood verification method.


than case 2020. Compared to the point-to-point method, the neighbourhood verification may significantly rise the skill score for scattered cases.

**Response:** Thank you for alerting us to this. This difference is the result of the different definitions of PAS and TS, where point-to-point comparisons are used in both the PAS and TS calculations and no neighbourhood verification methods are used.

Yes, compared to the point-to-point method, neighbourhood verification may significantly increase the skill score for scattered cases.

The typical case in Section 4.1 compares only the characteristics of PAS and TS, and in the newly added extreme precipitation case in the manuscript, the FSS scoring method was added for comparative experiments. For FSS verification, 15 km, 25 km, 45 km, 75 km and 120 km are used as the neighbourhood distances.

In the new case, the evaluation effect of FSS (45 km) for precipitation above 100 mm is better than that of TS . The evaluation feature of FSS is to examine the predictability scale of the model to reflect its predictive ability; however, due to the subjectivity of selecting neighbourhood scales, its score lacks comparability. In the new case, for small magnitude precipitation (above 0.1 mm and 10 mm) verification, the FSS scores tend to approach 1 as the neighbourhood distance expands, making it difficult to compare forecast differences between models. The results show that the PAS scoring method has obvious advantages in the evaluation of extreme precipitation events and can also reflect the differences in the small magnitude precipitation forecasting effects of the models well compared to those of the TS and FSS methods.

14. L441-443: the manuscript mentions "leading to a monotonous increase in scores" – what is it

referring to?

**Response:** Thank you for the comment. It refers to the magnitude-improved TS and the neighbourhood spatial verification method.

The modifications in the revised manuscript are as follows:

P22 L458-461: "*This result indicates that the PAS is different from the magnitude-improved TS and the neighbourhood spatial verification method. Both the magnitude-improved TS and the neighbourhood spatial verification method increase the tolerance, leading to a monotonous increase in scores.*"

This includes two situations: (1) the magnitude-improved TS score is higher than the original TS score (as explained in Table 2 in the introduction); and (2) the score of using the neighbourhood space verification method is higher than that of not using this method. This indicates that the PAS score is different from the magnitude-improved TS and the neighbourhood spatial verification methods, which monotonously increase the tolerance and monotonically increase the score. This also indicates that the PAS score has good recognition for extreme events.

15. L443-446: The statement, "The PAS assigns objective scores based on the proximity of the forecast to the observation, making it more reliable for precipitation evaluation than the TS," raises skepticism. Why is PAS considered more reliable than TS for precipitation evaluation? Is it because PAS is objective while TS is more subjective? If so, why expect the PAS evaluation align with the subjective judgment mentioned in L434-436?

**Response:** Thank you for the comment. We have changed the description in the revised manuscript as follows:

P22 L462-464: *"The PAS assigns scores based on the proximity of the forecast to the observation, making it more reliable for precipitation evaluation than the TS."*

The expert's questions are very insightful. Both the PAS and TS are objective evaluations based on certain rules.

PSS is more reliable than TS because TS has two natural flaws (threshold classification and the "double penalty" phenomenon). The threshold classification cannot guarantee that two adjacent precipitation values will always fall within the same threshold range. Slightly different precipitation values are not within the same threshold, which can lead to precipitation score distortion. The "double penalty" phenomenon leads to a score lower than the subjectively expected result, making it difficult to obtain appropriate verification scores when a forecast that "looks good" is not as good as one that "looks bad" (Ahijevych et al., 2009; Wilks, 2006; Ebert, 2008; Chen et al., 2021).

When comparing the advantages and disadvantages of the PAS and TS methods using actual cases, we first assume that the forecasts for these two cases can distinguish between good and bad forecasts.

This is a subjective judgement based on typical individual cases. When subjective judgement cannot distinguish which forecast is good, the conclusions drawn from PAS and TS are not easily accepted. For the two examples in this manuscript, the quality of the forecasts can be analyzed by the researchers themselves. We can analyze the reliability of the PAS and TS scores and then assess the quality of the scoring results based on such subjective judgements.

As explained by the article of Ahijevych et al. (2009), when verifying the effectiveness of scoring methods with real cases, it is also assumed that the predictions of these cases are good or bad and are determined through subjective scores. Of course, the author declares that "The panel's

subjective scores are alternative viewpoints, not definitive assessments of forecast performance."

Because each person's focus is different, for example, "Meteorologists are more likely to consider realistic depictions of meso-scale structure (such as in the stratiform precipitation area of a meso-scale convective system) as an important criterion defining a 'good' forecast, and may have focused on different features than scientists with a pure mathematical background." Therefore, the real case study is based on this assumption.

$\sum_n \text{PASN} = 24528$, $\sum_m \text{PAS}_{|ux0.1}= 10724.1788511084$;

$$\text{PASC} = \frac{1}{m+n}\left(\sum_m \text{PAS}_{|ux0.1} + \sum_n \text{PASN}\right) = \frac{1}{23472+24528}( 10724.1788511084 + 24528)$$

$$= 0.734420392731426$$

For details, see DataSoftware

/DataSoftware/1_Two_Typical_Processes/03Software_Configuration_Results_of_GCEM/Results_G

CEM folder in the outnc12hd22019071612.txt, outnc12hd22020061312.txt file is reflected,

screenshot as follow.

```
48076  n1=         23954 spasn=    23954.0000000000        nn1=        24046 spas=
48077    14828.6594345134
48078  pasc(spas+spasn)/(n1+nn1)=   0.807972071552363

48076  n1=         24528 spasn=    24528.0000000000        nn1=        23472 spas=
48077    10724.1788511084
48078  pasc(spas+spasn)/(n1+nn1)=   0.734420392731426
```

17. L473-474: For international readers' clarity, please mark the locations of Anhui, Zhejiang, Jiangxi,

Hunan, and Hebei on the figure.

**Response:** Thank the expert for this nice suggestion. Figure 10, as well as Figures 6, 7, 8 and 9, have been labeled with provincial administrative divisions.

18. The manuscript's English writing style makes it challenging to follow. Many expressions, such as "general comprehensive evaluation method" in the title, "core element of the precipitation forecast accuracy score index" in the abstract, and "the scoring areas in Zhejiang exhibit alternatively distributed high and low scores" in L467, sound awkward. Consider revising the manuscript for clarity and coherence.

**Response:** Thank the expert for the nice suggestions. For the "general comprehensive evaluation method", "general" means suitable for both general and extreme precipitation, while "comprehensive" means that the overall method includes several evaluation indices (methods).

The "core element of the precipitation forecast accuracy score index" was modified to the "*core indicator of the precipitation accuracy score (PAS)*" at P2 L30.

"The scoring areas in Zhejiang exhibit alternatively distributed high and low scores" was modified to "*the high and low scores in the Zhejiang region are scattered among them*" at P23 L495.

We have enhanced the English writing of the manuscript to ensure its clarity and coherence.

---

## Author Response (AR2)

Comments on the manuscript entitled "A General Comprehensive Evaluation Method for Cross-Scale Precipitation Forecasts" by Zhang et al. submitted to GMD

The authors have made revisions to the manuscript, including the addition of new sections and figures. However, I still have some minor comments for the authors' consideration:

1. L82: Please consider removing "shortly after".

**Response:** Thank you for alerting us to this. The statement has been modified as follows:

P4 L82-83: "*Gilbert (1984) proposed two scoring methods, namely, the ratio of verification and the ratio of success in forecasting.*"

2. L115-116: Could you clarify the intended meaning of "The reasons for which, although varied, are worthy of attention, but include its objectivity and practicality"?

**Response:** Thank you for alerting us to this. The statement is not accurate and has been modified as follows:

P6 L115-116: "*Although the reasons for this are varied, its objectivity and practicality merit attention.*"

The objectivity of the TS refers to the objectivity of its calculation method, while practicality refers to its simplicity, ease of use, and suitability for promotion. The sentence aims to convey that despite numerous factors contributing to the TS's consistently advantageous position, these two notable characteristics merit our attention. Consequently, in devising new scoring methods, it is essential to account for the advantages of the TS scoring method.

3. L127: Could you provide clarification regarding the spatial range of medium and small-scale

systems? Are you referring to "medium-scale" or "meso-scale"?

**Response:** Thank you for alerting us to this. The statement is not accurate, and it refers to the mesoscale, which has been changed from "modium" to "meso" in the text.

P6 L127: "meso- and small-scale".

P6 L129: "meso- and small-scale".

P7 L143: "meso- and small-scale".

P8 L173: "small- to meso-scale".

4. L135: Please elucidate the meaning of the sentence 'when a forecast that "looks good" is not as good as one that "looks bad"'.

**Response:** Thank the expert for the comment. With the development of high-resolution numerical weather forecasting, some meso- and small-scale phenomena have been portrayed by models. However, even if the spatial characteristics of forecasts are similar to those of observations, when there is a small deviation in the timing and location of events between a forecast and an observation, both "false alarms" and "missed alarms" will occur, leading to a score lower than the subjectively expected result.

As an example mentioned in Ahijevych et al. (2009) (Fig. 1), for the comparison of geom001 and geom005 predictions, although the geom001 forecast is consistent with the observation in terms of morphology, there is a deviation of displacement, and no overlap between the forecast and observation, then the TS score is 0. On the other hand, the geom005 forecast and observation are quite different, but due to the overlap between the forecast and observation, the TS score is 0.11. Of course, the article also points out that even if modelers and other users believe that geom005

predictions are very poor, a hydrologist might actually prefer geom005.

In conclusion, a larger TS value does not necessarily indicate a better overall forecast. There will be situations where the forecast looks good but the TS score is poor.

[Figure]

**Figure 1.** (a)–(f) Five simple geometric cases derived to illustrate specific forecast errors. The forecasted feature (red) is positioned to the right of the observed feature (green). Note, in (f) (geom005), the forecast and observation features overlap. One grid box is approximately 4 km on a side.

References:

Ahijevych, D., Gilleland, E., Brown, B. G., and Ebert, E. E.: Application of spatial verification methods to idealized and NWP-gridded precipitation forecasts, Weather Forecast., 24, 1485–1497, https://doi.org/10.1175/2009WAF2222298.1, 2009.

5. L159-163: Could you specify the context in which you are referring to "using the traditional skill score"?

**Response:** Thank the expert for the comment. Simple upscaling method adjusts the high-resolution forecast and observation information to a larger scale, and then using the traditional skill score. Here "using the traditional skill score" specifically refers to the threat score (TS), and also includes

equitable threat score (ETS), Frequency Bias (BIAS), False Alarm Ratio (FAR), Missing Ratio (MR),etc.

6. L198-199: What is meant by "base rate independence"?

**Response:** Thank the expert for the comment. The base rate is a descriptive statistical term used in categorical events for deterministic forecasting, representing the probability of an observed event occurring. The base rate, alternatively referred to as observational probability or climatological probability, serves as a fundamental descriptive index. It is not a performance indicator. Independent of the base rate is an ideal attribute of scoring indicators, because the scoring does not change with climate change (Yule, 1912). The designed indicators should be suitable for both the rainy and dry seasons within the same region. Also they can be applied to various regions, including arid and semi-arid areas, as well as semi-humid and humid areas.

The concept of "base rate independence" in the manuscript, which is not clearly expressed in the current context, has been changed to "independent of climatological probability" to enhance readability.

P9-10 L198-200: "*The designed scoring performance indices should possess ideal attributes such as fairness, independent of climatological probability, suitability for extreme events, and boundedness as much as possible.*"

References:

Yule, G. U.: On the methods of measuring association between two attributes, Journal of the Royal Statistical Society, 75, 579-652, 1912.

7. L198-199: What is meant by "The devised scoring method should be easy to promote"?

**Response:** Thank the expert for the comment. "The devised scoring method should be easy to promote" means "The new scoring method should be conceptually clear, pragmatic, and operational, with broad applicability across various regions and seasons."

8. L216-221: Please consider rephrasing the lengthy sentence for better readability.

**Response:** Thank the expert for the suggestions. The statement has been modified as follows:

P10-11 L216-221: "*To address the issues of "distorted scores due to the division of precipitation thresholds and increased subjective risks brought about by the setting of the neighbourhood spatial verification method" in traditional and improved precipitation scoring methods, this study refers to the verification method for heavy rainfall forecasts based on predictability (Chen et al., 2019) and combines the advantages of relative and absolute errors. A GCEM is constructed by directly analyzing the proximity of forecasted precipitation to observed precipitation.*"

9. L240, Eq.3: The authors argue that EPI indicates the excessive precipitation index. Thus, u should not be equal to x. However, I disagree. EPI equals 0 when u=x, indicating no excessive precipitation. Including the case u=x in Eq.3 might be beneficial. Assuming one wants to evaluate precipitation forecast using observed precipitation data with missing values, the EPI will be missing in locations/times where no observation is available. EPI is also missing when u=x if Eq. 3 does not include the case of u=x. How do you propose to distinguish between these cases when EPI is a missing value?

**Response:** Thank the expert for the comment. Through the expert analysis, we have learned that the expert have different perspectives on our definition of EPI. Indeed, as the expert has pointed out, in the current situation (u ≠ x), when EPI is missing, it cannot be determined whether it is caused by u=x or because u itself is missing. In fact, in the current situation, when EPI is missing, it can only indicate cases that do not belong to (0<u<x), which may include cases where u is missing, u=x, or u>x. If Eq. 3 is set to u=x, then Eq. 2 should also be set to u=x. This will result in duplicate calculations, where u=x needs to be calculated both in EPI and in IPI. Another reason is that we want to calculate the degree of over forecasting and under forecasting when there are biased forecasts. Therefore, we do not define u=x within the calculation range of EPI or IPI.

10. L246, 250: Eq. 4 combines Eqs. 2 and 3. However, Eqs. 2 and 3 do not include u=x, whereas Eq. 4 does. Additionally, I still believe it may not be necessary to define IPI and EPI. It seems analogous to already having mean error; hence, defining a positive mean error index and a negative mean error index may be unnecessary. However, I am open to retaining IPI and EPI if the authors find it necessary. Nevertheless, removing IPI and EPI could help in sharpening the focus of the paper.

**Response:** We extend our gratitude to the expert for agreeing to retain IPI and EPI in the case of different opinions.

11. Figs. 1-4: The font size of the x- and y-axes is too small, especially for Fig. 3. Please consider using a larger font size.

**Response:** Thank you for alerting us to this. Figures 1-5 have been adjusted appropriately.

12. Code and data: I suggest saving the source code and data separately instead of combining them into one large file. This would facilitate easier downloading of the source code. Additionally, the data should be saved in individual files rather than being divided into parts within a single RAR file. This current setup requires users to download all files before the data can be uncompressed, which could be inconvenient.

**Response:** Thank the expert for the suggestions. The code and data have been stored separately, and the data is stored in individual files.